# Synoptic weather patterns conducive to compound extreme rainfall-wave events in the NW Mediterranean

Marc Sanuy[1], Juan C. Peña[2], Sotiris Assimenidis[2], José A. Jiménez[1]

[1]Laboratori d'Enginyeria Marítima, Universitat Politècnica de Catalunya, BarcelonaTech, c/Jordi Girona 1-3, Campus Nord ed. D1, Barcelona, 08034, Spain
[2]Servei Meteorològic de Catalunya, C. Berlin, 38-46, 08029, Barcelona, Spain

*Correspondence to*: Marc Sanuy (marc.sanuy@upc.edu)

**Abstract.** The NW Mediterranean coast is highly susceptible to the impacts of extreme rainstorms and coastal storms, often leading to flash floods, coastal erosion, and flooding across a highly urbanised territory. Often, these storms occur simultaneously, resulting in compound events that intensify local impacts when they happen in the same location or spread impacts across the territory when they occur in different areas. These multivariate and spatially compound events present significant challenges for risk management, potentially overwhelming emergency services. In this study, we analysed the prevailing atmospheric conditions during various types of extreme episodes, aiming to create the first classification of synoptic weather patterns (SWPs) conducive to compound events involving heavy rainfall and storm waves in the Spanish NW Mediterranean. To achieve this, we developed a methodological framework that combines an objective synoptic classification method based on principal component analysis and k-means clustering with a Bayesian Network. This methodology was applied to a dataset comprising 562 storm events recorded over 30 years, including 112 compound events. First, we used the framework to determine the optimal combination of domain size, classification variables, and the number of clusters, based on the synoptic skill to replicate local-scale values of daily rainfall and significant wave height. Subsequently, we identify SWPs associated with extreme compound events, often characterized by upper-level lows and trough structures in conjunction with Mediterranean cyclones, resulting in severe to extreme coastal storms combined with convective systems. The obtained classification demonstrated strong skill, with scores exceeding 0.4 when considering factors like seasonality or the North Atlantic Oscillation. These findings contribute to a broader understanding of compound terrestrial-maritime extreme events in the study area and have the potential to aid in the development of effective risk management strategies.

**Keywords:** Bayesian Network, synoptic skill, objective weather classification, multivariate events, spatially compound events, NAO, seasonality, coastal storms, flash floods, cut-off, Mediterranean cyclones.

## 1. Introduction

The NW Mediterranean coast frequently experiences flash floods during heavy rainfall episodes (Llasat and Puigcerver, 1992; Llasat et al., 2014; Gaume et al., 2016) and coastal storms when high waves erode and flood the coastal fringe (Mendoza et al., 2011; Jiménez et al., 2012). When these hazards coincide, they result in compound extreme events that can have significant impact regionally. These compound events increase the local impact where both hazards concur at the same basin and can also distribute the impact widely across the territory when they occur simultaneously in different areas (Zscheischler et al., 2020). For instance, the recent Gloria storm in January 2020 had substantial repercussions in Catalonia (NW Mediterranean, Spain). This storm led to four casualties, structural damage in different breakwaters along the coast, a railway bridge collapse, and impacts associated with extreme waves, rainfall, and wind action distributed throughout the territory (Canals and Miranda, 2020). The Spanish national reinsurance company, Consorcio de Compensación de Seguros, paid €143 million for storm-induced damage, limited to insured property (Luján-López, 2022). Managing the risk associated with such events is challenging due to the intensity of the observed damages and their extensive spatial coverage, which can overwhelm protection services (e.g. during the Gloria storm, civil protection services received approximately 15,000 calls and performed 2510 services in 4 days, as reported by the Ministry of Home Affairs of the Government of Catalonia).

Compound coastal floods have gained increasing attention in recent years (Wahl et al., 2015; Wu et al., 2018; Bevacqua et al., 2019; Hendry et al., 2019; Camus et al., 2022). However, despite their high relevance in the NW Mediterranean, these events have predominantly been studied within large-scale (continental) analyses of the multivariate type, where different hazards act simultaneously at the same location (Paprotny et al., 2018; Zscheischler et al., 2018, Bevacqua et al., 2019). A recent analysis of compound events in the NW Mediterranean found that they occur relatively frequently, approximately 3.4 times per year, and are often spatially compound, involving the simultaneous occurrence of heavy rainfall and storm waves in different parts of the region, while some few areas concentrate multivariate events (Sanuy et al., 2021). Assessing the magnitude of the impact and associated risks involves considering the intensity of the involved hazards, and their distribution over the territory, to be combined with the level of exposure and vulnerability of affected areas (e.g. Kron, 2013).

Since these hazards are influenced by specific weather conditions, identifying meteorological patterns conducive to extreme events becomes crucial. This knowledge can indirectly estimate their probability of occurrence (Catto and Dowdy, 2021; Camus et al., 2022) and can therefore become a relevant element in risk management. Various methods have been used for weather classification, ranging from manual procedures (Hess and Brezowky, 1969; Lamb, 1972) and correlation-based map typing (Yarnal, 1993; Wu et al., 2018) to more objective approaches such as principal components and clustering (e.g. Huth et al., 2008). The EU COST 733 project (Huth et al., 2010) has significantly contributed to advancing scalable classification techniques applicable to various European regions. Several classification methodologies were proposed and rigorously compared within this project, highlighting that different classification approaches demonstrated comparable effectiveness (e.g., Philipp et al., 2010, Beck and Philipp, 2010). Notably, the synoptic skill of weather classifications, i.e. their capacity to accurately replicate the magnitudes of key target variables at the local scale, was identified as particularly sensitive to various methodological aspects inherent to objective approaches. These encompassed factors such as the predefined number of classification groups, the selection of atmospheric variables and their number, and the spatial dimensions of the classification domain (see e.g. Philipp et al., 2016; Beck et al., 2016; Teegavarapu et al., 2018; Falkena et al., 2020). In this context, Bayesian Networks (BNs) have been employed to assess the dependencies between multiple non-linearly related variables in natural hazard analysis (Beuzen et al., 2018; Plant et al., 2016; Couasnon et al., 2018) and can therefore be used to evaluate the synoptic skill of objective weather classification methods in reproducing local-scale variability in target variables. Additionally, BNs provide a framework for assessing conditioned probabilities between weather patterns, their climatological factors, and their characteristics at local scale (e.g. Sanuy et al., 2021).

While weather patterns dominant during individual rainfall and coastal storms in the NW Mediterranean have been well identified (Martín-Vide et al., 2008; Mendoza et al., 2011; Gilabert and Llasat, 2017; Gil-Guirado et al., 2021), synoptic weather patterns (SWPs) conducive to compound events involving both hazards in the region have not been thoroughly studied. Recent analysis of concurrent intense precipitation and extreme wind events over the Iberian Peninsula identified their link with cyclones and associated atmospheric rivers; however, the results suggested that the climatology of extreme weather events in the NE sectors (NW Mediterranean coast) is driven by different mechanisms, relative to other sectors (Hénin et al., 2021). Mediterranean cyclones have been highlighted as significant contributors to extreme precipitation and coastal storm impacts (Flaounas et al., 2022). Inter-annual variations in wave height during coastal storms are predominantly influenced by seasonality (Morales-Márquez et al., 2020), while those of extreme precipitation are more closely tied to the North Atlantic Oscillation (NAO) (Casanueva et al., 2014; Krichak et al., 2014; Criado-Aldanueva and Soto-Navarro, 2020). The complex orography of the region influences precipitation and wind patterns, and the coastal storm climate is characterised by the waves playing a more significant role to erosion and flooding than surges (Mendoza and Jiménez, 2009; Sanuy et al., 2020).

To investigate the meteorological conditions under which compound events involving heavy rainfall and storm waves occur in the NW Mediterranean and understand their impact on contributing hazards and how they vary across the territory, we classified weather conditions during recorded storm events, both univariate and compound, spanning 30 years (1990-2020). We utilized Principal Component Analysis (PCA) in combination with k-means clustering as an objective classification approach, considering multiple combinations of atmospheric variables, domains and number of clusters. The selected atmospheric variables included mean sea level pressure (mslp), 500 hPa geopotential height (z500), 850hPa temperature (t850) and 10m wind fields (u,v). Initially, the BN prediction skill is used to identify the most suitable combination of domain, variables and clusters number of groups for individual and compound events. Subsequently, we relate the obtained patterns to storm characteristics across the region by calculating BN conditioned probabilities. This study represents the first classification and characterization of weather patterns conducive to compound events on the NW Mediterranean coast.

## 2. Study area and data

### 2.1. Study area

The study area encompasses the approximately 600 km long coastline of Catalonia in the NW Mediterranean, including its river basins (Figure 1). The presence of many small torrential catchments combined with the orographic forcing on Mediterranean air masses, contributes to convective instability, altering pressure fields and causing intense rainfall and flash floods (Jansà et al., 2014), Hydrological and morphological factors have been shown to significantly influence flash flood occurrences (Llasat and Puigcerver, 1992; Llasat et al., 2014). Coastal storms in the area are dominated by NE–E extreme waves, with secondary impacts from S–SE (Mendoza et al., 2011). Coastal flooding primarily results from wave-induced run-up, while storm surges play a secondary role in inundation in this part of the NW Mediterranean (Mendoza and Jiménez, 2009; Sanuy et al., 2020). Over recent decades, the region has experienced a shoreline retreat due to reduced sediment supply from rivers and the presence of numerous obstacles disrupting natural littoral dynamics (Jiménez and Valdemoro, 2019). Consequently, wave action on a progressively narrowing coastline has led to a significant increase in coastal damages (Jiménez et al., 2012).

The terrestrial part of the study area was divided into seven sectors of natural river catchments along the coast (hereafter called basins), based on the literature (Llasat et al., 2016; Sanuy et al., 2021). This subdivision allowed for a focused analysis of direct pluvial hazards occurring in the lower part of the basins, excluding the upper part of long and regulated river catchments, such as the Ter in Girona and the Llobregat in Barcelona (Figure 1). Moreover, the marine part was divided into three large sectors covering the main wave climate areas of the coast: north, central, and south. Each wave sector characterised the corresponding waves in more than one basin (Figure 1).

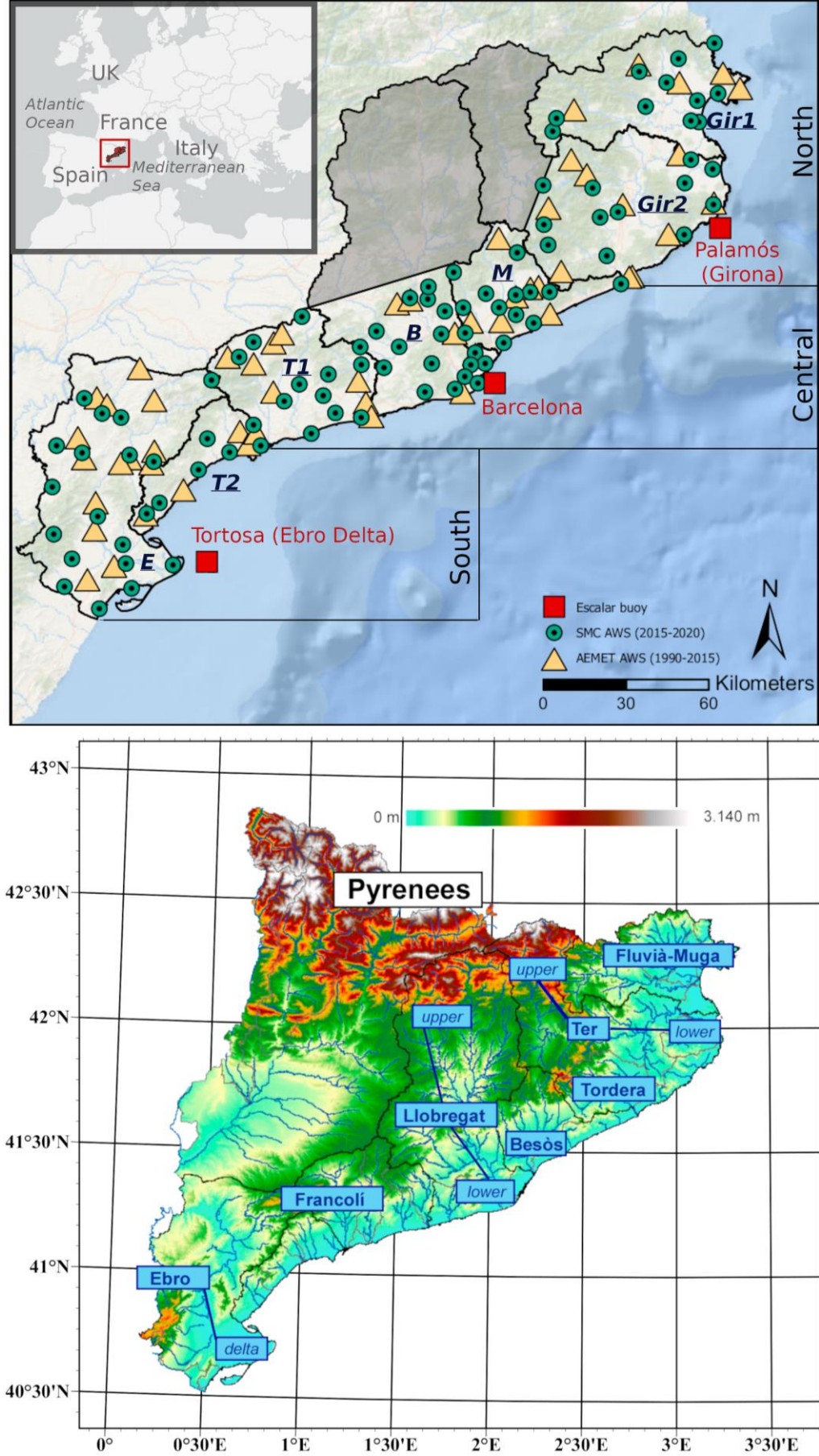

**Figure 1. Top: Location of the study area, and map showing the distribution of dataset nodes in river basins and coastal sectors. Rainfall data were recorded at Automatic Weather Stations -AWS- (circles and squares), while wave data were recorded by three wave buoys (red squares). Bottom: Topography and main rivers at the study site. Maps created with ESRI-ArcGIS base maps and elevation data provided by Institud Cartogràfic i Geológic de Catalunya.**

**2.2. Data**

 **2.2.1 Rain**

Rainfall data for the study area were collected by Automatic Weather Stations (AWS) from two networks distributed throughout the study area, covering the coastal basins of interest (Figure 1, Table 1). The datasets included daily cumulative rainfall for 1990-2015 recorded by the Spanish State Meteorology Agency (AEMET) and 30 min data for 2015-2020 by the Meteorological Service of Catalonia (SMC), which were converted to daily cumulative precipitation before merging.


**Table 1. Number of Automatic Weather Stations (AWS) per basin and dataset.**

| Costal Sector | Basin | | AEMET AWS (1990-2015) | SMC AWS (2015-2020) |
|---|---|---|---|---|
| North | Gir1 | (Girona North) | 6 | 11 |
| | Gir2 | (Girona South) | 9 | 12 |
| Central | M | (Maresme) | 7 | 13 |
| | B | (Barcelona) | 5 | 16 |
| | T1 | (Tarragona North) | 8 | 12 |
| South | T2 | (Tarragona South) | 5 | 7 |
| | E | (Ebro Delta) | 14 | 18 |
| Total | | | 54 | 86 |

**2.2.2 Waves**

The wave data consisted of a reconstruction of the significant wave height (Hs) and peak period (Tp) for the Palamós,
Barcelona, and Tortosa wave buoys (Figure 1). The reconstruction process was based on ERA5 data (SMC, 2021) and utilized a multilinear regression technique, employing five oceanic variables (significant wave height, total wave mean period, mean wave period based on the first moment, mean zero-crossing wave period and total wave peak period) and three atmospheric variables (mean sea level pressure, wind speed and wind direction at 10m) as predictors for the targeted buoy variables (Hs or Tp). To account for the influence of wind and the morphology of the Catalan coast, the data was categorized into four groups
based on wind direction (0º to 90º, 90º to 180º, 180º to 270º, and 270º to 360º), resulting in distinct regression coefficients for each group.

**2.2.3 Atmospheric variables**

The synoptic classifications were based on ERA5 data (Hersbach et al., 2023a and 2023b), comprising various combinations
of meteorological parameters, including mslp, z500, t850, and both components of 10m wind (u and v). This data span from 1990-2020 and cover a synoptic scale extending from 25ºW to 30ºE longitude and 30ºN to 65ºN latitude, with a spatial resolution of 0.25ºx0.25º and a temporal resolution of 3 h.

Monthly mean NAO time series data for the same period were obtained from the Climate Prediction Center (CPC) NOAA website (http://www.cpc.noaa.gov/data/, last accessed in July 2023).

**3. Methods**

**3.1. General framework**

The developed methodology consists of three main steps (Figure 2):

- (A) Event identification: This step involves identifying and characterizing rain and wave storm events as individual or compound events, based on the presence of either or both meteo-marine drivers.

- (B) Weather classifications: Here, synoptic weather patterns were identified by using a PCA and k-means approach, utilizing multiple atmospheric variables, domains, and varying cluster sizes.
  - (C) BN analysis: In this step, a BN is employed to select an appropriate combination of domain, classification variables and number of clusters (N). The BN was also employed to identify critical SWPs, and to characterize the probabilistic spatial distribution of intensity and local simultaneity of compound events.


To evaluate the synoptic classifications, different domains and variable combinations were defined, as their skill is sensitive to these factors (Beck et al., 2016). The largest domain (D0) corresponds to the downloaded synoptic scale. Additional regions were defined (Figure 3, Table 2), and the synoptic skill in estimating basin-scale wave heights and daily rainfalls across these domains were compared. The k-means approach involved various combinations of atmospheric variables (Table 2), including

the use of each individual variable as a predictor, as well as groups of 2 to 4 selected from those that individually performed the best. Consequently, Step B (Section 3.3) and the BN skill analysis (Section 3.4) were conducted for each of the three event types identified in Step A. Furthermore, this analysis was performed for each combination of domain, k-means variable set, and number of clusters (Table 2).

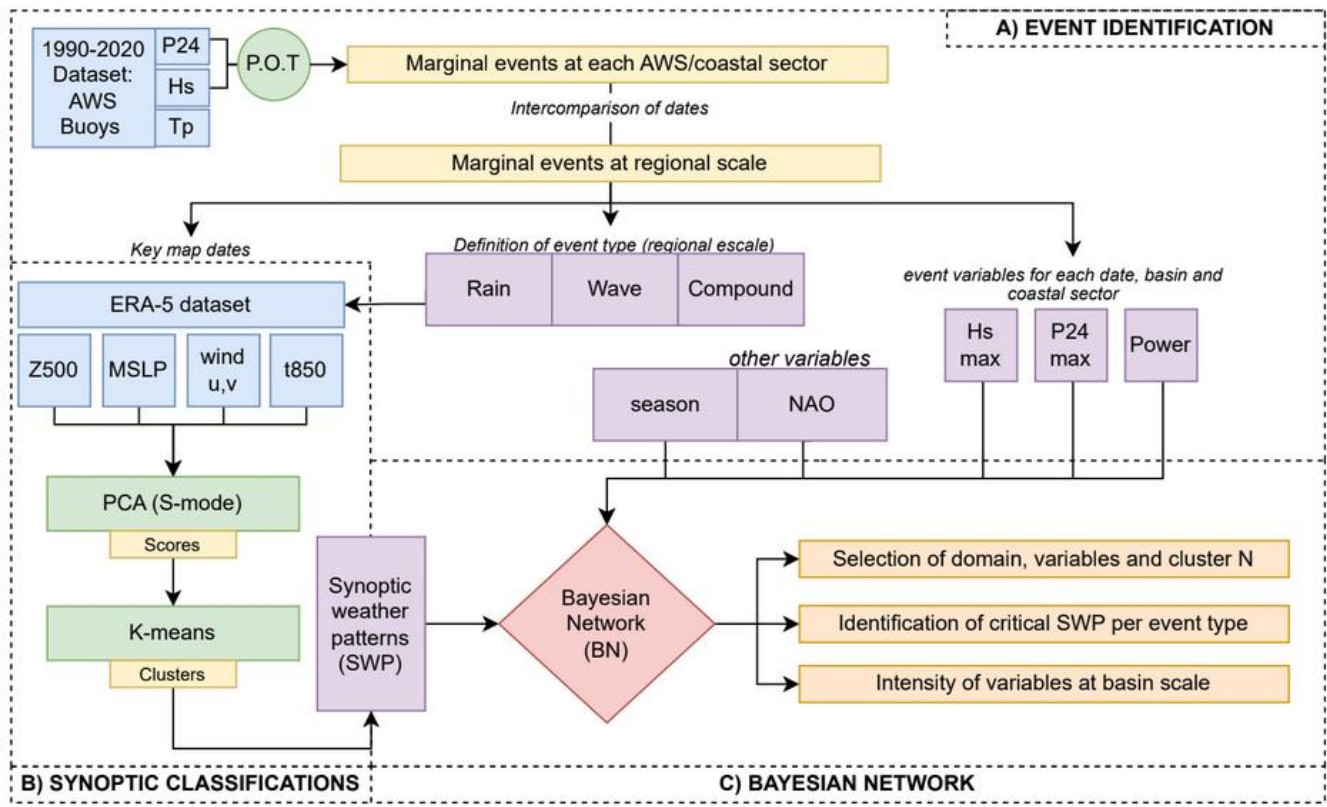

**Figure 2. General methodological framework. A) Both compound and individual events are identified and characterized based on**
**their peak conditions. B) The PCA + k-means approach is applied to both compound and individual events, with varying combinations of domain sizes, classification variables, and the number of clusters. C) The Bayesian Network is used to assess the synoptic skill in estimating P24 and Hs at the local scale. Once the appropriate combination is determined, the BN is used to identify critical SWP based on their intensity and the spatial distribution of extreme conditions at local scale.**

### 3.2. Event identification

In each basin, rain and coastal storm individual events were identified using the Peak Over Threshold (POT) method for all stations within the basin. Rainstorms were identified with a P24h threshold value of 40 mm and a 3-day interval to establish independent events. This threshold has been identified as the lower limit for events that could potentially lead to flash floods if they are locally convective (Barbería et al., 2014; Cortès et al., 2018). For each event in a basin, the maximum P24 at any station and the accumulated P24 at that station were retained, resulting in a single time series of rainstorms per basin. Wave

storms were identified using a dual-threshold approach (Sanuy et al., 2020) with a 0.98 quantile, a minimum duration of 6 hours, and 3-day lag to identify independent events and define their duration. The second threshold, set at the 0.995 quantile, was used to retain only extreme episodes. The absolute thresholds for the 0.995/0.98 quantiles, aggregating data from the three wave datasets, were 2.9/2.2 m, respectively. Each event was characterised by its maximum Hs and power content, approximated as the integration of $H^2*Tp$ over the storm duration. Also, the season and the monthly average NAO

corresponding to each event was saved to be later used in the BN analysis.

The final dataset of storms included three event types in regional scale: (i) compound events, with presence of both rain and wave events at any of the basins; (ii) rain events, characterised by a rainstorm of P24 ≥ 40 mm in at least one basin without the co-occurrence of a wave storm in any coastal sector; and (iii) wave events, characterised by the absence of a rainfall event

in any basin but with a wave storm in any of the coastal sectors. Within the compound events, further classification at the basin scale was done, distinguishing between multivariate events (both rain and waves in the same basin), compounding wave events (basin affected only by waves), or compounding rain events (basin affected only by rain). Sanuy et al. (2021) emphasized the importance of this classification proposed by Zscheischler et al (2020) for risk management during compound events in NW Mediterranean conditions.


Storm events, both individual and compound, underwent basic exploratory analysis to determine their annual and seasonal frequencies and average magnitudes.

### 3.3. SWP classification

To identify weather patterns, we employed an objective methodology previously used in the area by Aran et al. (2011) and

Gil-Guirado et al. (2021), consisting of two main steps: PCA and Cluster Analysis (CA). We apply the he PCA + CA approach separately to rain, wave, and compound events identified in the previous step, using the +00 UTC maps corresponding to each event's peak (maximum P24 for the rain events and maximum Hs for wave and compound events). This classification procedure was applied across all 7 domains.

**Table 2. Number of PCAs included in the k-means approach per variable, event type and domain. Combinations of variables and number of clusters tested per event type and domain.**

| Domain | | | PCAs rain / waves / compound | | | | | k-means variable combinations | k-means number of clusters (N) |
|---|---|---|---|---|---|---|---|---|---|
| id | lon | lat | mslp | z500 | u | v | t850 | **Individual:** | **Rain:** |
| D0 | [-25 , 30] | [30 , 65] | 40 / 22 / 27 | 21 / 19 / 23 | 152 / 42 / 61 | 84 / 28 / 42 | 98 / 50 / 66 | mslp, z500, u, v, t850 | [14, 18, 26] |
| D1 | [-15 , 25] | [30 , 55] | 31 / 16 / 21 | 14 / 14 / 16 | 84 / 25 / 38 | 87 / 27 / 41 | 60 / 42 / 51 | **Combined:** | |
| D2 | [-15 , 15] | [30 , 55] | 26 / 14 / 18 | 11 / 12 / 14 | 66 / 20 / 31 | 73 / 23 / 37 | 53 / 40 / 49 | {mslp+u}, {msl+z500} | **Waves:** |
| D3 | [-10 , 10] | [30 , 55] | 22 / 11 / 15 | 09 / 09 / 11 | 60 / 19 / 29 | 66 / 22 / 33 | 41 / 36 / 42 | {z500+u}, | [6, 10] |
| D4 | [-15 , 25] | [35 , 50] | 21 / 12 / 16 | 10 / 11 / 13 | 76 / 22 / 36 | 78 / 23 / 38 | 45 / 37 / 43 | {mslp+z500+u}, | |
| D5 | [-15 , 15] | [35 , 50] | 18 / 11 / 13 | 08 / 09 / 11 | 54 / 16 / 28 | 60 / 19 / 31 | 39 / 35 / 41 | {mslp+z500+u+v} | **Compound:** |
| D6 | [-10 , 10] | [35 , 50] | 15 / 09 / 11 | 06 / 07 / 09 | 46 / 14 / 25 | 53 / 16 / 27 | 30 / 31 / 35 | {mslp+z500+u+t850} | [6 , 10 , 14] |

PCA was used to reduce the dimensionality of mslp, z500, t850 and wind (u,v) data within each domain (Table 2). Additionally, we applied PCA to the standardized anomalies of the maps, separately for each variable. The determination of the number of

principal components aimed for 99% explained variance for variables with low variability (mslp, z500, t850) and 90% for those with high variability (u and v) (Table 2). Subsequently, CA was employed on the PCA score matrix to identify the SWPs. We utilized the non-hierarchical k-means method (McQueen, 1967) and tested various combinations of atmospheric variables

and different number of clusters (N) as outlined in Table 2.  Classifications yielding clusters with fewer than 5 members (dates) were considered not robust.

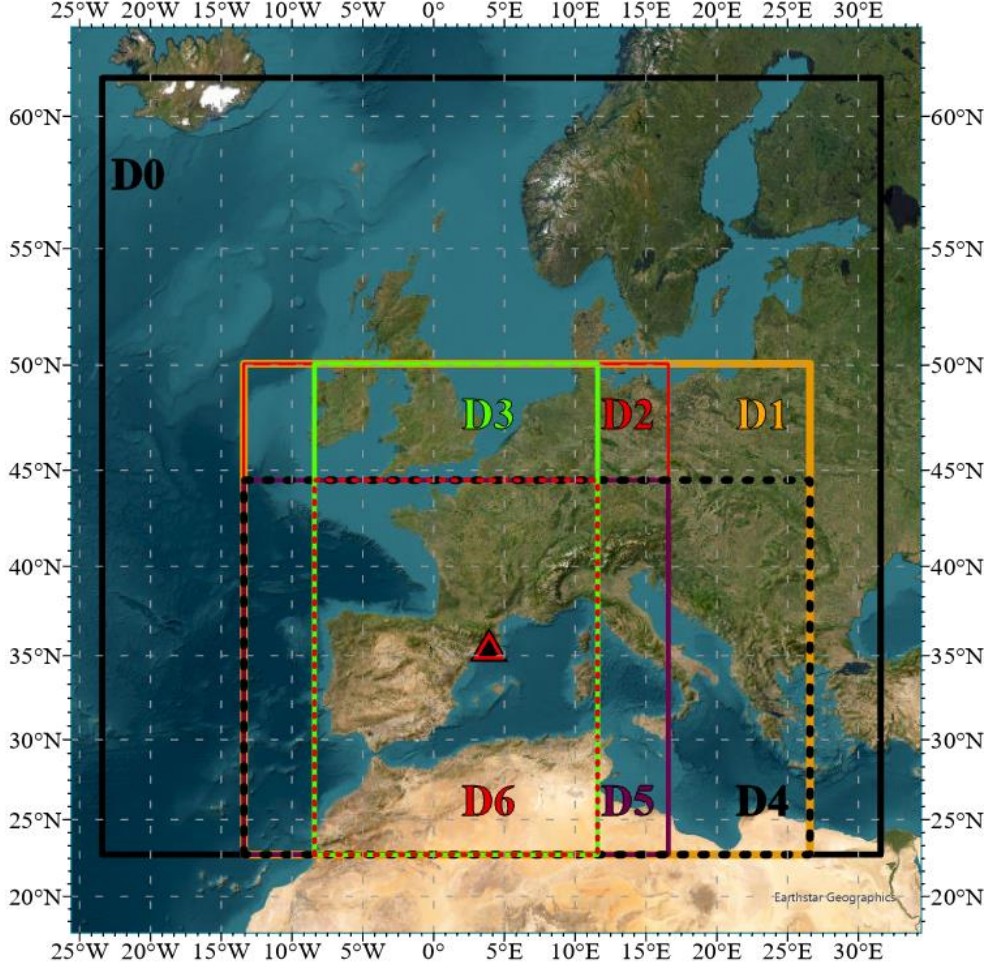

**Figure 3. Domain sizes tested in the classification analysis. Triangle indicates the location of the study area.**

### 3.4. Bayesian Network

BNs are tools used to analyse complex systems by assessing the interdependence and conditional probability of variables (Pearl, 1988; Jensen, 1996). They have been effective in descriptive and predictive applications for natural hazards, which are different in how the variables are connected and discretised (Beuzen et al., 2018). Descriptive BNs were used to see through the nonlinear relationships between variables in order to explore specific relationships between predictors and target variables (Figure 4).

The BN variables correspond to the essential elements required for evaluating the performance of classifications (SWP) in estimating target variables (Hs and P24) and characterizing coastal storm intensities (Power) on a basin scale (Area), also when they are combined with seasonality or the NAO. The discretisation of the variables was determined based on expert knowledge, identifying threshold combinations representative of the study site.

Regarding hazard intensity, P24 = 40 mm represents a local threshold for flash flood potential due to convective events (Barbería et al., 2014; Cortès et al., 2018), while P24 = 100 mm and P24 = 200 mm are the thresholds for potentially damaging and extreme rainstorms, respectively, according to the warning criteria of the Meteorological Service of Catalonia (as defined on www.meteocat.cat, last accessed April 2023). The variable P24 was additionally discretised using uniform steps between the thresholds. Wave storms were classified using the local classification system of Mendoza et al. (2011), which uses storm power content (*Power*). Class II (moderate) events are characterized by a mean Power = 3500 $m^2$hs and peak Hs ≥3 m, while

class IV (severe) storms exhibit a mean Power = 9100 $m^2hs$ and mean peak Hs = 5 m. Energy content class values from Mendoza et al. (2011) were translated into power content thresholds by applying a factor of 10, assuming a representative mean value of Tp of 10s throughout the entire duration of wave events. We also considered the warning criteria established by the Meteorological Service of Catalonia for coastal storms, which relies solely on significant wave height (Hs). According to this criterion, hazardous conditions are categorized as "high level" when Hs reaches or exceeds a threshold of 4 meters. In this study, we classified the results into three intensity levels: (i) Extreme conditions that comprise individual or compound events with P24 ≥ 200 mm, storm power content ≥ IV, and Hs ≥ 4m; (ii) Severe conditions that encompass events that do not meet the extreme criteria but have P24 ≥ 100 mm, a classification of ≥ II for storm power content, and Hs ≥ 4m; and Moderate conditions that include events exceeding POT thresholds but fall short of meeting the severe criteria.

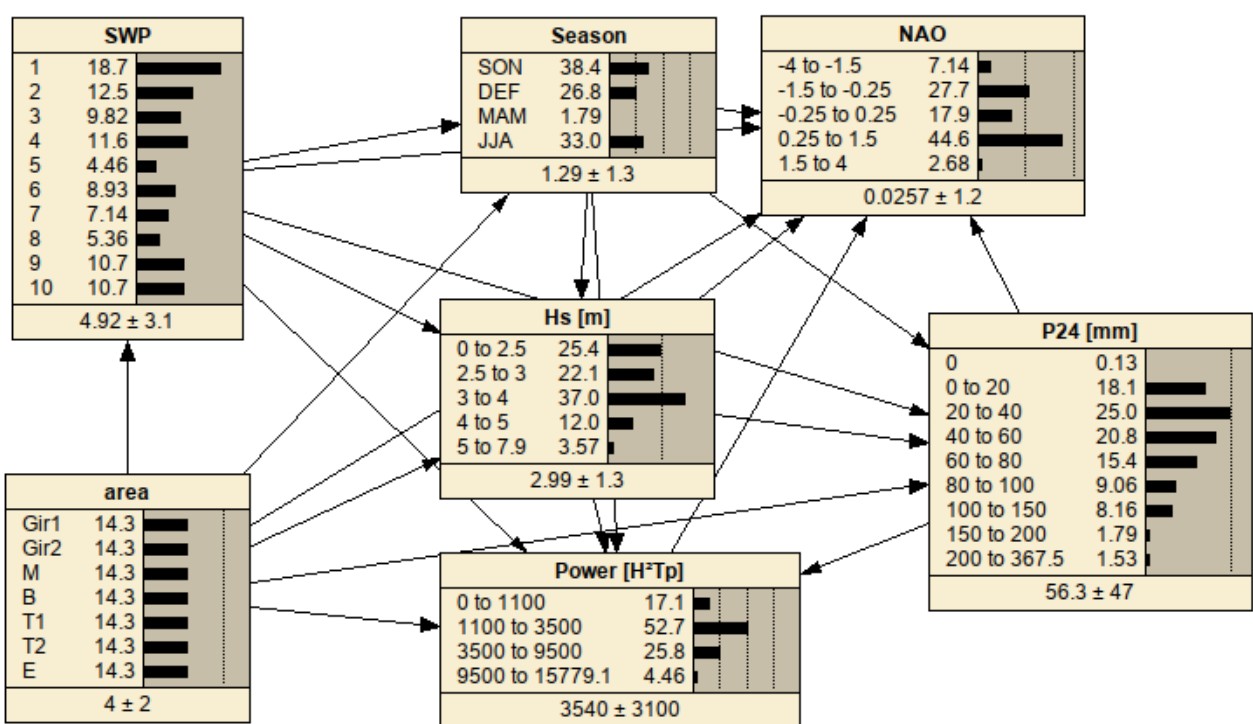

**Figure 4. Example of Bayesian Network (BN) used evaluate classifications and analyse SWPs. This example corresponds to the BN for N=10 groups, trained with the k-means classification using z500 and u at D0.**

We generated five BNs to test different values of N (i.e. 6, 10, 14, 18, 26), and created a training dataset for each combination of event type, domain, and N, resulting in a total of 88 tests. Each training dataset included one vector of variables for each event and basin, totalling 2,632 training vectors for rain events, and 518/784 cases for wave/compound events, respectively. To assess the SWP system's predictive capability for estimating Hs and P24h, known as synoptic skill, we calculated the BN-skill measure as *skill = 1-msr/msd,* where *mrs* is the mean square of the residuals of the linear regression between the mean value of the BN-predicted probability distributions and the data, and *msd* is the mean square of the data. The standard deviation of the prediction can be used as a weight, giving less weight to uncertain predictions (high standard deviation) and more weight to confident outputs (Plant et al., 2016). Consequently, an incorrect but uncertain prediction is then considered more skilful than a confidently incorrect prediction. Both weighted and unweighted skills were calculated for the BN tests of interest, using SWP alone or in combination with seasonality or the NAO as predictors. In meteorological applications, skill values above 0.2 are generally considered "useful," those surpassing 0.4 are categorized as "good", and when they achieve or exceed 0.6, they are regarded as "excellent" (Sutherland et al., 2004).

The BN was also used to calculate conditioned probabilities, particularly for assessing the probability of SWP occurrence at predefined intensity levels. BN training incorporates the values of variables for each basin and event, resulting in presentation probabilities that encapsulate both the temporal frequency of SWPs and the number of basins impacted at specified intensity thresholds. Furthermore, we derived probabilities for intensity levels at the basin scale, conditioned by SWP, and calculated the probability of the variable "Area" conditioned by multivariate occurrences of such intensity levels.

## 4. Results

### 4.1. Compound events characterisation

From 1990 to 2020, the study area experienced 112 compound episodes, of which 61 were classified as severe in at least one component (i.e. with $P24 \geq 100$ mm or $Hs \geq 4m$ with storm class $\geq$ II). Regarding the other two event types, 376 rain and 74 wave episodes were recorded during the same period, of which 37 rain events were severe, and only 3 of the wave events reached a class II storm with $Hs \geq 4m$. On average, the entire territory experienced 3.7 compound events/year, 16.3 rain events/year, and 6.2 wave events/year. The frequency of severe storms in these types averaged at 2, 1.23, and 0.1, respectively. It is worth noting that no statistically significant trend was detected in any event category during the analysis period (Figure 5).

The seasonal distribution conditions contributing to compound events exhibit significant differences. Rainstorms occur year-round, with two peaks occurring in spring and late summer-autumn. By contrast, wave storms were notably absent during the summer months, with their main season spanning from September to April (Figure 5). The combination of these distinct temporal patterns conditions the occurrence of compound events year-round, which reflects the temporal pattern of wave storms, which—although less frequent than rainstorms—are the controlling factor. One significant property of these compound events, in contrast to individual storms, is their potential for greater damage. This is evident in the fact that when heavy rain and wave storms coincide, the characteristics that determine their intensity, such as rainfall, wave height, and storm power, are magnified, along with a larger spatial extent (as indicated by the number of affected basins) compared to when they occur individually (Figure 5). During the wave storm season, the maximum accumulated precipitation values at a single station were, on average, 55**%** higher during compound events than during standalone rainstorms, while maximum Hs and wave power were on average 4**%** and 20 **%** higher, respectively, than during individual wave storms.

Despite the relatively small study area, which comprises approximately 600 km of coastline, the occurrence and characteristics of these events display regional variations due to the influence of the orography and coastal configuration conditioning them. Notably, the annual frequency of compound events was highest in the north, averaging 2.6 events per year, compared to 1.2 events per year in the central and southern coastal sectors. It is important to mention that the occurrence of a compound multivariate event affecting simultaneously the entire area is "rare" (Sanuy et al., 2021) and has only happened during some extreme storms (e.g. Gloria). This extreme storm, along with other four episodes, constitute just 0.9% of the total events and 4.5% of the compound events.

This spatial variability was also reflected in the magnitude of the events. The wave storm energy flux was fairly evenly distributed across the sectors, with the largest Hs observed in the north. Similarly, rainstorms presented higher average P24 values in the northern sector. These results align with findings from the BN analysis of SWP, as discussed in section 4.3.

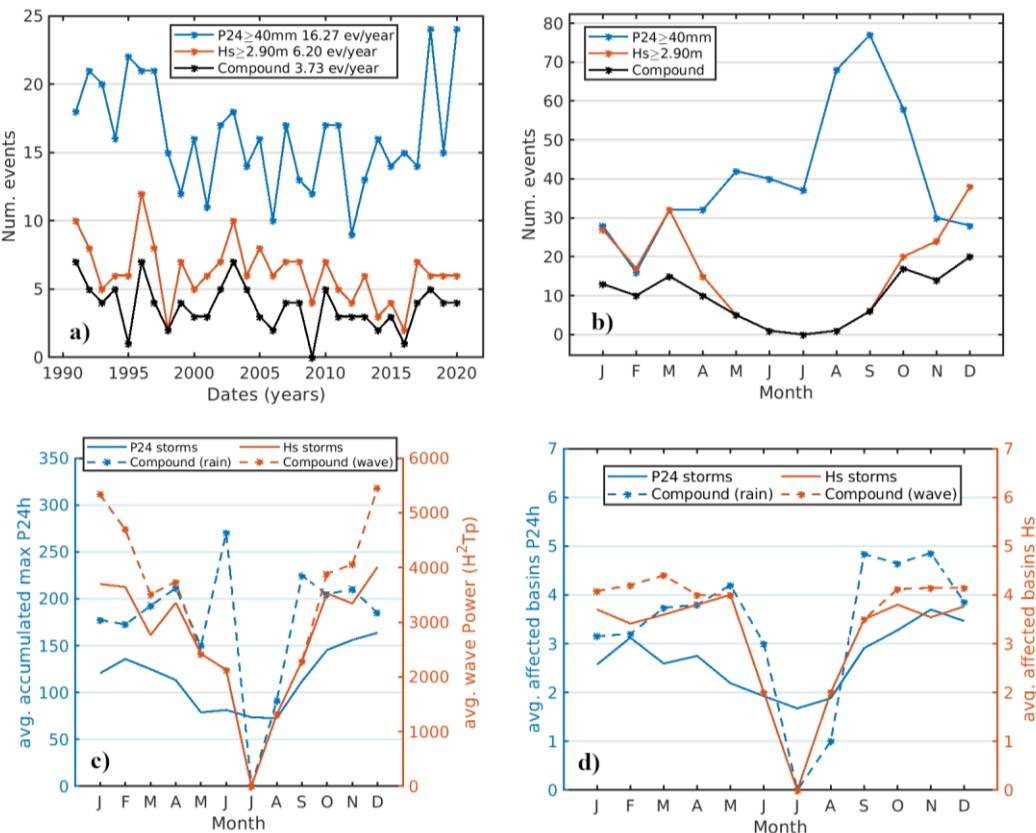

Figure 5. (a) Annual occurrence of event types. (b) Monthly occurrence of event types during the period of study. (c) Monthly average magnitudes of cumulative rainfall and wave power during individual and compound events. (d) Monthly average number of affected basins during individual and compound events along the Catalonian coast.

**4.2. Selection of domain, classification variables and number of clusters**

As previously mentioned, we employed the BN to guide the selection of the domain for characterizing synoptic weather types, the choice of variables, and the determination of the optimal number of clusters for classifying these weather types, all of which were based on the BN skill assessment (see section 3.4). The results of the BN skill assessment, including the identification of clusters with fewer than 5 events (referred to as "bad clusters"), are presented in Supplementary Figures S1, S2, and S3. A summary of the results is presented in Table 3, where the best-performing classifications in groups containing 5 or more dates are highlighted.

For compound events, skills were calculated as the average between Hs and P24 skills, while for wave and rain events, individual target variables were used to determine their respective skills. Notably, the most promising classification was observed for N=10, where z500 and u were used as classification variables at domain D0. This configuration yielded skills ranging from 0.24 (SWP only) to 0.45 (SWP and NAO combined). Conversely, all classifications with N=14 resulted in groups containing a limited number of dates, as depicted in Figure S1.

Regarding wave types, only one instance of N=10 clusters met the requirement of having at least 5 dates per cluster, utilizing mslp and u at D0. This, yielded skills ranging from 0.32 (SWP alone) to 0.51 (SWP + NAO). Interestingly, the skills of SWP alone or SWP + NAO were comparable for N=6 and N=10, while SWP + season skills saw the most benefit at N=10, reaching a skill value of 0.47, closely approaching the performance of SWP + NAO for compound events.

In the case of rain events, none of the combinations for N up to 26 achieved a skill level of 0.2 when relying solely on SWP. Acceptable skills were obtained when SWP was combined with season, starting at N=14. Particularly, for N=18, skills were

nearly equivalent. The best results were observed within acceptable clusters for N=26, utilizing the combination of z500 and u at D6, which yielded a skill of 0.33 when incorporating SWP with NAO as predictors.

An overarching analysis of the entire test set (see Figures S1, S2, and S3) reveals that, unlike rain and wave events, the highest skills for the compound type were achieved with the largest domain. Furthermore, combining two or more variables consistently led to enhanced skills compared to single-variable models, especially when applied at larger domains and for increasing values of N.

Notably, the weighted skills, accounting for BN output uncertainty, spanned a range from 0.5 (using only SWP) to 0.9 (using SWP in conjunction with NAO). This underscores the BN's capacity to effectively capture uncertainty in its estimations. Specifically, when the BN output deviated significantly from the measurements, the BN output distribution exhibited a larger standard deviation. Conversely, instances when the BN output exhibited a low standard deviation closely aligned with the measurements. This analysis highlights the potential for improved prediction accuracy by introducing additional variables into the BN framework.

**Table 3. Summary of top skills within acceptable classifications (i.e. excluding classifications yielding clusters with fewer than 5 days) for each event type. The optimal combination, based on BN skill, for each event type is highlighted in bold.**

| Event type | Domain | N | variables | skill (swp) | skill (swp+season) | skill (swp+NAO) |
|---|---|---|---|---|---|---|
| **Compound** | 0 | 6 | mslp, z500 | 0,2 | 0,29 | 0,31 |
| Avg (Hs,P24) | 0 | | mslp, z500, u, v | 0,2 | 0,3 | 0,33 |
| | 0 | | z500 | 0,2 | 0,28 | 0,36 |
| | 1 | | u | 0,2 | 0,33 | 0,32 |
| | 0 | **10** | mslp, z500, u, v | 0,22 | 0,33 | 0,37 |
| | 0 | | z500 | 0,2 | 0,31 | 0,4 |
| | **0** | | **z500, u** | **0,24** | **0,33** | **0,45** |
| **Wave** | 0 | 6 | mslp, z500, u | 0,29 | 0,38 | 0,48 |
| (Hs) | 2 | | mslp | 0,32 | 0,34 | 0,49 |
| | 4 | | mslp, z500, u, t850 | 0,3 | 0,35 | 0,45 |
| | **0** | **10** | **msl,u** | **0,32** | **0,47** | **0,51** |
| **Rain** | 1 | 14 | mslp, u | 0,13 | 0,2 | 0,24 |
| (P24) | 4 | | mslp, z500, u | 0,12 | 0,2 | 0,24 |
| | 4 | | u | 0,12 | 0,2 | 0,21 |
| | 6 | | mslp, z500, u | 0,12 | 0,21 | 0,25 |
| | 0 | 18 | mslp, z500, u, t850 | 0,1 | 0,21 | 0,22 |
| | 1 | | z500 | 0,12 | 0,2 | 0,26 |
| | 1 | | z500, u | 0,11 | 0,2 | 0,26 |
| | 2 | | z500 | 0,12 | 0,2 | 0,26 |
| | 0 | **26** | z500, u | 0,11 | 0,25 | 0,31 |
| | 2 | | z500 | 0,13 | 0,23 | 0,31 |
| | **6** | | **z500, u** | **0,12** | **0,24** | **0,33** |

For illustrative purposes, we conducted an additional test using the best classification of compound events, namely z500 and u at D0 with 10 clusters (Figure 6). In this test, we incorporated both the NAO and the season predictors, together with the SWP. However, to prevent overfitting and manage the complexity of the model, we constrain the NAO variable to three bins: positive NAO>0.25, negative NAO<-0.25 and approximately neutral NAO $\in$ [-0.25, 0.25]. The estimation of P24 was carried out in two scenarios: one with the node linked to Hs and Power (Figure 6, a), and another without this link (Figure 6, b).

Notably, allowing the BN to account for relationships within SWP between rain and wave intensity directly impacted the output uncertainty, as illustrated in Figure PS. The P24 skill improved to 0.49 when the nodes were linked, compared to a skill of 0.43 without the link. Interestingly, the weighted skills reached 0.95 in both cases, indicating that the advantage of noise reduction was captured by the conventional measure of skill. Therefore, the test demonstrates that incorporating knowledge about Hs and storm energy content can enhance the prediction of P24, as these variables exhibit partial correlations.

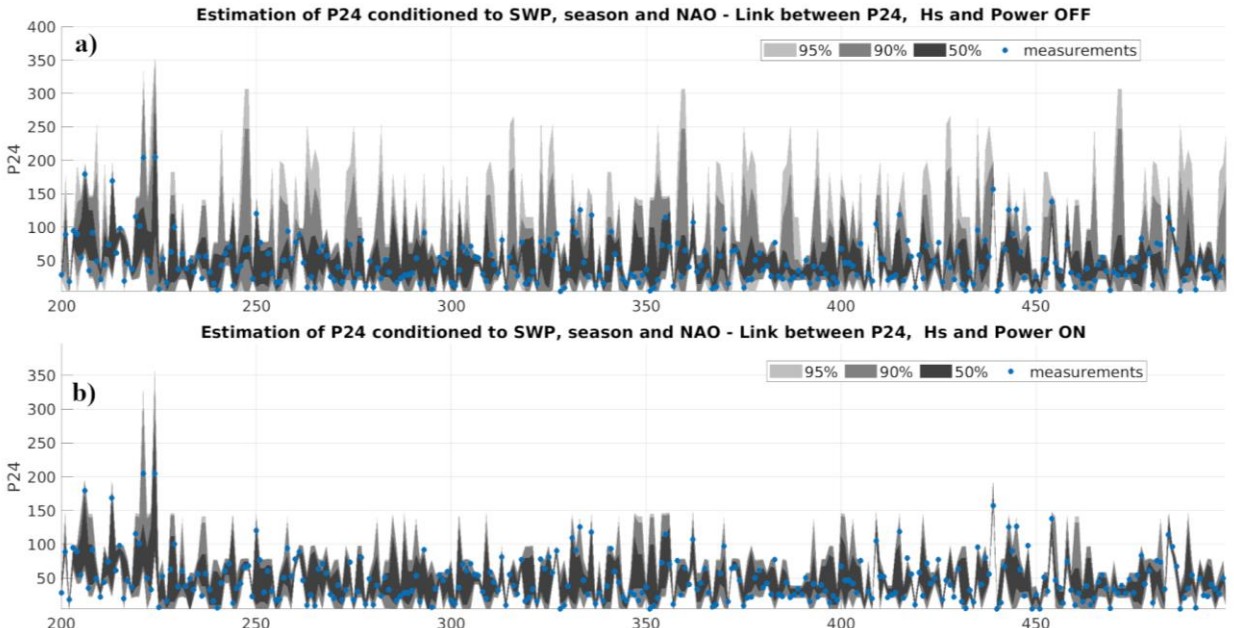

**Figure 6. Test results for z500 and u at D0 using SWP, NAO and season as predictors. The BN has to estimate 784 cases, corresponding to 112 events at 7 basins. For illustration purposes, cases 200 to 500 are shown (x axis). The blue dots represent the measurements, while the shaded areas depict the BN output distribution. (a) Test without a link between target variables P24, Hs and Power. (b) Test with the target variables linked.**

### 4.3. Synoptic weather patterns conducive to extreme compound events

The compound SWPs, derived from the combination of z500 and u with N = 10 clusters, can be classified based on their conditional presentation probabilities concerning the various intensity levels of the key variables, P24 and Hs (see Section 3.4). The corresponding probabilities, obtained through the trained BN, are presented in Table 4.

An initial set of patterns corresponds to extreme conditions in both rainfall and coastal storms, specifically, P(SWP|P24≥200mm) and P(SWP|C≥IV & Hs≥4m). These patterns, denoted as SWP1, SPW4, and SPW5, collectively account for 75% of extreme daily rainfall and 78.8% of extreme coastal storms during compound events in the study area (Table 4).

**Table 4. Probability of occurrence of each SWP, unconditioned (in blue) and conditioned on different intensities of the interest variables (in red). The colour intensities vary from maximum to minimum values within each column.**

| SWP | N dates | P(SWP) | 1 - P(SWP\|P24≥200mm) | 2 - P(SWP\|P24≥100mm) | 3 – P(SWP\|C≥IV & Hs ≥ 4m) | 4 – P(SWP\|C≥II & Hs ≥ 4m) | 1+3 Multivariate | 2+4 Multivariate |
|---|---|---|---|---|---|---|---|---|
| 1 | 21 | 18,8 % | 41,7 % | 25,6 % | 51,5 % | 32,1 % | 66,7 % | 50,0 % |
| 4 | 13 | 11,6 % | 25,0 % | 11,1 % | 12,1 % | 26,4 % | 33,3 % | 16,7 % |
| 5 | 5 | 4,5 % | 8,3 % | 12,2 % | 15,2 % | 4,7 % | 0,0 % | 12,5 % |
| 7 | 8 | 7,1 % | 16,7 % | 12,2 % | 0,0 % | 0,0 % | 0,0 % | 0,0 % |
| 2 | 14 | 12,5 % | 8,3 % | 8,9 % | 0,0 % | 8,5 % | 0,0 % | 4,2 % |
| 6 | 10 | 8,9 % | 0,0 % | 8,9 % | 15,2 % | 14,2 % | 0,0 % | 4,2 % |
| 9 | 12 | 10,7 % | 0,0 % | 5,6 % | 6,1 % | 7,5 % | 0,0 % | 12,5 % |
| 10 | 12 | 10,71 % | 0,00 % | 10,00 % | 0,00 % | 3,77 % | 0,00 % | 0,00 % |
| 8 | 6 | 5,36 % | 0,00 % | 4,44 % | 0,00 % | 0,00 % | 0,00 % | 0,00 % |
| 3 | 11 | 9,82 % | 0,00 % | 1,11 % | 0,00 % | 2,83 % | 0,00 % | 0,00 % |

SWP1 exhibits the highest probability of occurrence when conditioned on extreme rain and wave intensities, with probabilities reaching 42% and 52%, respectively (Table 4). This suggests that approximately half of the instances involving compound

events featuring at least one extreme driver in any basin can be attributed to this specific pattern. It is worth noting that SWP1 exclusively corresponds to class IV coastal storms and multivariate severe events across all 7 basins (see Figure 7-c, d). Furthermore, the prevalence of P24≥100mm is most pronounced in the northernmost sectors, followed by Barcelona and the Ebro Delta. Notably, SWP1 accounts for 67% of multivariate extreme occurrences and 50% of multivariate severe events at the basin scale (as indicated in Table 4). This pattern is characterized by a cut-off structure at the z500 level, accompanied by a high-pressure system centred over the British Isles that extends eastward and southward. Additionally, a Mediterranean cyclone is situated south or southwest of the Iberian Peninsula (see Figure 7-a, b).

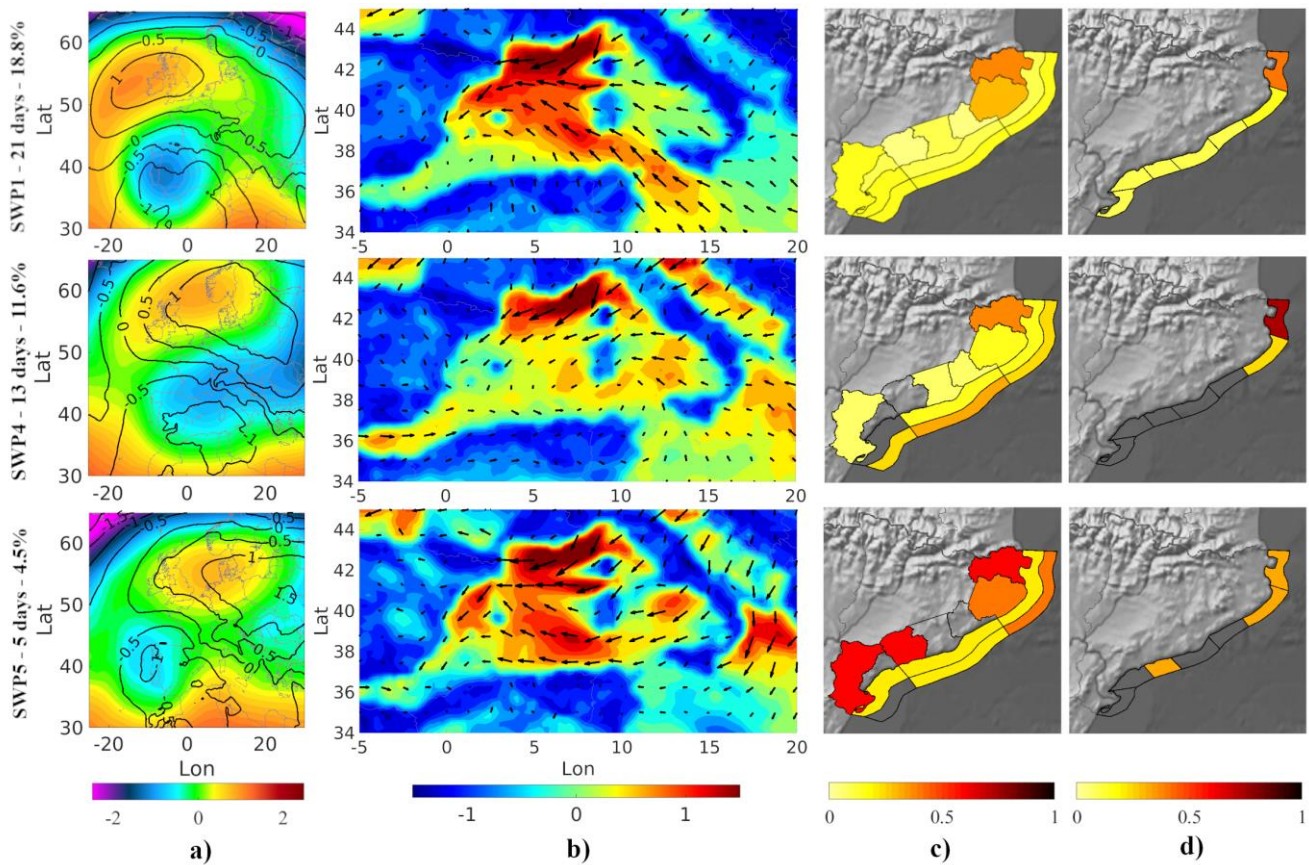

**Figure 7. Compound SWP associated with P24>200mm and coastal storms of C≥IV and Hs ≥ 4 m. (a) z500 anomaly (colormap) overlaid with mslp anomaly (contours). (b) Wind anomaly. (c) Probability of P24≥100 mm (land) at each basin; probability of storm power content of class ≥ IV (internal coastal band) and probability of Hs ≥ 4m (external coastal band) at each coastal sector. (d) Spatial distribution of multivariate severe events of P24≥100mm with a coastal storm of class ≥ II and Hs ≥ 4m.**

This is followed by SWP4 and SWP5 in accordance with intensity (Table 4). SWP4 exhibits a stronger association with P24≥200mm (25%), whereas SWP5 is more closely linked with class IV storms featuring Hs over 4m (15.2%). SWP4 is characterized by an upper-level trough extending from eastern Europe to the Iberian Peninsula, coupled with a high-pressure system over the North Sea. At lower levels, high pressures extend toward eastern Europe, accompanied by a Mediterranean cyclone located near Sardinia (see Figure 7-a, b). SWP4 tends to generate class ≥ 4 coastal storms in the northern and central coastal sectors, with P24≥100mm mainly concentrated in the north, where severe multivariate episodes occur. In contrast, central and southern basins tend to experience compounding waves or rainfall, with a higher likelihood of P24≥100mm occurring in Maresme, Barcelona, and the Ebro Delta. On the other hand, SWP5 is characterized by an upper-level low pressure system over Portugal, coupled with a Mediterranean low situated near Algiers (see Figure 7-a, b). The distribution of drivers across the territory differs from that of SWP4, with class 4 storms reaching the Ebro Delta, and severe rainfalls being more probable at the southern third of the study area. SWP5 also tends to concentrate Hs ≥ 4 m more to the north compared to SWP4.

Moving on, a second set of patterns, represented by SWP7 and SWP2, is linked to extreme levels of rainfall, accounting for 16.7% and 8.3% respectively (Table 4). However, these patterns involve moderate to severe coastal storms. Both SWP7 and SWP2 share common characteristics with Mediterranean low-pressure systems located either over the Balearic Islands or to the west. Additionally, they exhibit the presence of a trough originating from the British Isles.

In the case of SWP7, this trough is accompanied by high-pressure systems extending towards eastern Europe (see Figure 8-a, b). Conversely, SWP2 exhibits a similar structure, but it tends towards a cut-off pattern over the Iberian Peninsula, featuring a broader trough extending eastward. These structural differences influence the distribution of meteorological drivers across the territory. SWP7 leads to severe rainfall events across all basins, with higher probabilities in the southern third of the region. However, it does not result in class II storms with Hs ≥ 4m, thereby precluding the occurrence of multivariate episodes. On the other hand, SWP2 affects the northern and southern regions, albeit with lower probabilities. In this case, it triggers class II coastal storms with Hs ≥ 4m, leading to the observation of multivariate events in the northern areas (Figure 8-c, d).

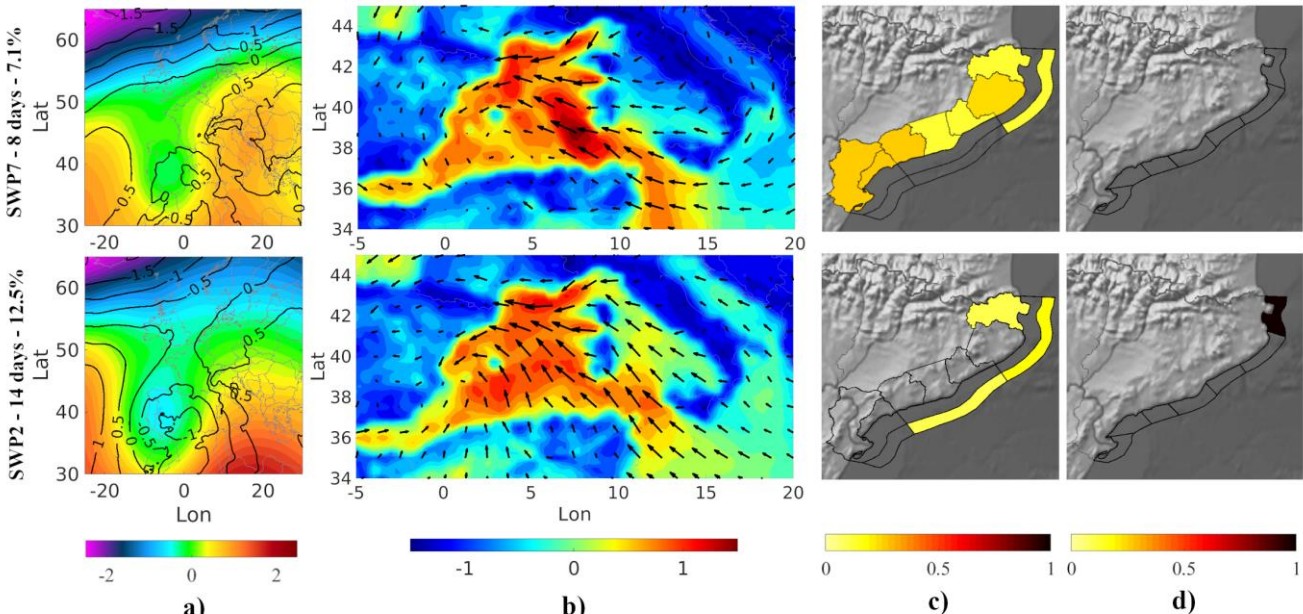

**Figure 8. Compound SWP associated with P24>200mm and coastal storms of II ≤ C ≤IV with Hs ≥ 4 m. (a) z500 anomaly (colormap) overlaid with mslp anomaly (contours). (b) Wind anomaly. (c) Probability of P24≥100 mm (land) at each basin; probability of storm power content of class ≥ IV (internal coastal band) and probability of Hs ≥ 4m (external coastal band) at each coastal sector. (d) Spatial distribution of multivariate severe events of P24≥100mm with a coastal storm of class ≥ II and Hs ≥ 4m.**

A third group of patterns exhibits distinctive characteristics, involving moderate to severe rainfall coupled with extreme coastal storms. Notably, SWP6 is linked to 15.6% of extreme coastal storms across various sectors and contributes to 4.2% of severe multivariate events in different basins (Table 4). In the case of SWP6, both the cut-off pattern and the Mediterranean low are centred over the Balearic Islands, accompanied by a high-pressure system located over northern Europe (see Figure 9-a, b). This atmospheric configuration creates conditions conducive to extreme coastal storms and severe rainfall events spanning from north to south, with Hs≥4m primarily concentrated in the central coastal sectors. However, multivariate events are observed exclusively in the northernmost basin. Moving on to SWP9, although it has a slightly lower impact on extreme waves (6.1%), it exhibits a higher association with multivariate severe events (12.5%). SWP9 is characterized by an Atlantic low situated close to the northwest of the Iberian Peninsula, alongside a high-pressure system over Scandinavia. Notably, there is no presence of a Mediterranean low (Figure 9-a, b). This atmospheric configuration results in coastal storms originating from the southern direction, affecting the central and northern sectors with class ≥ IV storms, accompanied by Hs ≥ 4m, severe rainfall, and multivariate episodes at the northern regions (Figure 9-c, d).

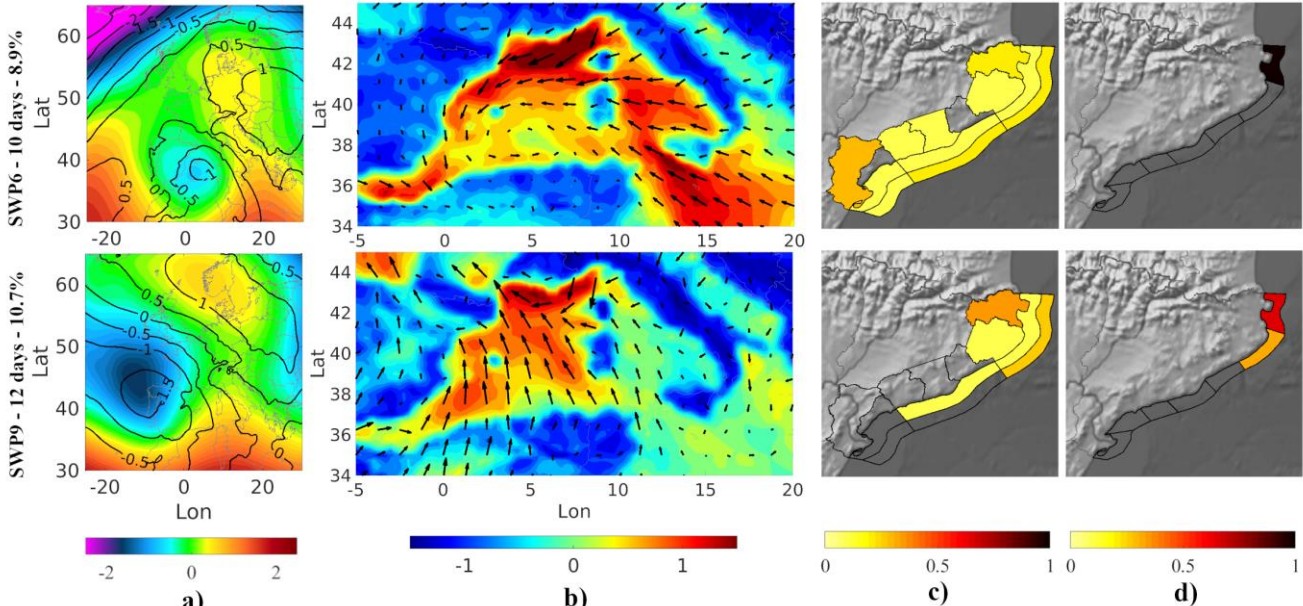

**Figure 9. Compound SWP associated with 100 mm ≤ P24 ≤ 200 mm and coastal storms of C≥IV and Hs ≥ 4 m. (a) z500 anomaly (colormap) overlaid with mslp anomaly (contours). (b) Wind anomaly. (c) Probability of P24≥100 mm (land) at each basin; probability of storm power content of class ≥ IV (internal coastal band) and probability of Hs ≥ 4m (external coastal band) at each coastal sector. (d) Spatial distribution of multivariate severe events of P24≥100mm with coastal storm of class ≥ II and Hs ≥ 4m.**

The remaining patterns, namely SWP10, SWP8, and SWP3 (Figure 10), correspond to events characterized as moderate to severe in both rain and wave components. Notably, there are no instances of severe multivariate episodes across any of the basins (Table 4). In the case of SWP10, it is marked by a north-western Atlantic trough with a Mediterranean low positioned over the Balearic Islands, albeit with lesser intensity compared to previous patterns (see Figure 10-a, b). This atmospheric configuration leads to P24≥100 mm across the entire territory, accompanied by waves reaching Hs ≥ 4m and class II coastal storms primarily in the northern sector (Figure 10-c). SWP8 is characterized by a cut-off pattern over the Iberian Peninsula, with a corresponding low-pressure system located over the Gulf of Cádiz (Figure 10-a, b). While winds in the Mediterranean do not generate severe coastal storms in the study area, severe rainfall events may occur in Barcelona and the northern basins (Figure 10-c). Finally, SWP3 is marked by a Mediterranean low situated between the Gulf of Lyon and the Ligurian Sea, accompanied by a configuration of low-pressure systems at both at the surface and upper levels, extending from the British Isles to the North Sea and Scandinavia (see Figure 10-a, b). This atmospheric setup results in winds blowing from the land toward the Mediterranean Spanish coast, with Hs reaching 4m at specific locations in the central and northern sectors. Severe rainfall events were recorded exclusively in the northernmost basin Figure 10-c).

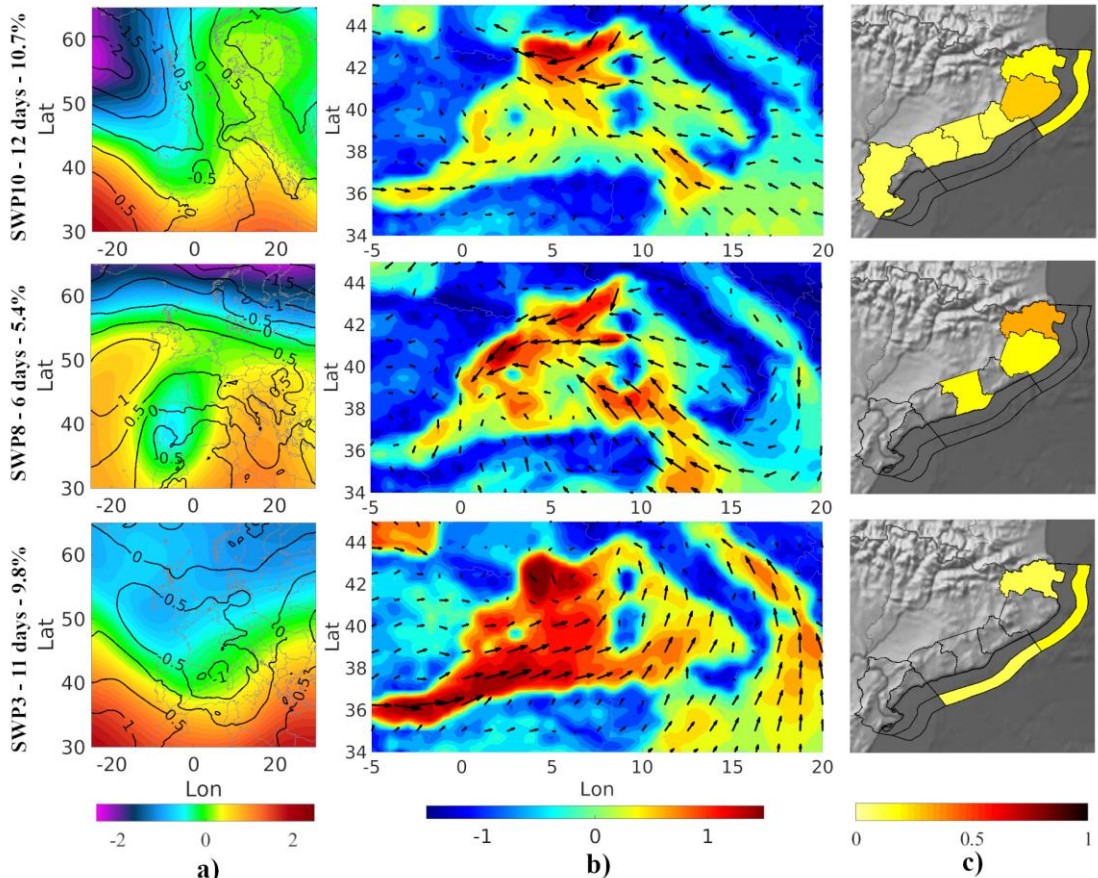

**Figure 10. Compound SWP associated with 100 mm ≤ P24 ≤ 200 mm coastal storms of II ≤ C ≤IV with Hs ≥ 4 m. (a) z500 anomaly (colormap) overlaid with mslp anomaly (contours). (b) Wind anomaly. (c) Probability of P24≥100 mm (land) at each basin; probability of storm power content of class ≥ IV (internal coastal band) and probability of Hs ≥ 4m (external coastal band) at each coastal sector. Please note that none of the SWPs were linked to severe multivariate episodes; therefore, the panels for such cases are not displayed.**

## 5. Discussion

Our integrated framework, which combines an objective classification method with Bayesian Networks, has proven effective in identifying and characterizing extreme compound SWPs. Notably, the majority of SWPs associated with extreme conditions exhibited upper-level lows or trough formations, often in conjunction with Mediterranean lows or cyclones. These atmospheric patterns resulted in severe to extreme coastal storms combined with convective systems. These identified patterns align closely with impactful events previously documented in the literature (see Table 5). The descriptions of their impact closely correspond to the probabilistic characterizations derived from the BN framework. In contrast, none of the synoptic patterns associated with extreme individual rain or wave events featured this specific combination of atmospheric structures (Supplementary Figures S4, S5). For instance, the strong winds responsible for individual wave events did not originate from Mediterranean lows and were not associated with extreme coastal storms, typically reaching only the severe intensity level (Figure S5). Similarly, extreme individual rain episodes were linked to cut-off and trough configurations without the presence of a Mediterranean cyclone (Figure S4).

These results are consistent with previous individual SWP classifications of extreme rainfall (Romero et al., 1999, Gázquez et al., 2004), convective rainfall (Barbería et al., 2014; Cortès et al., 2018), floods (Gil-Guirado et al., 2021), coastal storms (Mendoza et al., 2011), and strong winds (Peña et al., 2011) in the study area. For instance, the extreme compound SWP1 (Figure 7) was identified by Gil-Guirado et al. (2021) as the configuration most associated with terrestrial floods on the Spanish Mediterranean coast. This pattern was also recognized by Peña et al. (2011), who linked the cut-off configuration to the

presence of strong winds, heavy showers, and sea wave erosion in the region. Regarding marine hazards, Mendoza et al. (2011)

conducted a preliminary analysis of synoptic situations associated with severe to extreme coastal storms (class III–V). They found that the most extreme storms (class V events) occurred in the presence of a Mediterranean cyclone. Configurations without Mediterranean lows, referred to as Southern and Eastern Advections, were also associated with less severe coastal storms ranging from class II to IV.

**Table 5. Compilation of significant historical events associated with the identified extreme compound SWP, with concise summaries**

**of documented impacts along with corresponding references.**

| SWP | Date | Description | References |
|---|---|---|---|
| SWP1 | 20/22-1-2020 | Record-breaking events at all basins with accumulated precipitations exceeding 500 mm. Exceptional flooding with casualties at the Ter and Tordera rivers. Extensive damages due to the extreme coastal storm. | Amores et al. (2020) Canals & Miranda (2020) Sanuy et al. (2021) |
| | 13-10-2010 | More than 100 mm in the northern basins causing flash floods | ACA (2011) |
| | 11/15-11-2001 | Record winds of 170 km/h in northern basins. Strong coastal erosion over the whole territory. Double peak coastal storm with rainfalls during the second peak | Genovés & Jansá, (2002) Mendoza et al. (2011) |
| | 30-1-2006 | The rainstorm caused several villages and houses to be cut off due to rivers and streams overflowing. Sixty-five municipalities in the Girona North basin were left without electricity supply for almost six hours due to the repair work on the high-voltage tower that was knocked when the Fluvià river burst its banks. | ACA (2011) |
| | 7/8-5-2002 | Severe and extreme rainfalls were recorded at all basins. A reservoir in Girona North is forced to open because it has reached 87% of its capacity. | ACA (2011) |
| | 12-11-1999 | Generalised rainfall with P24 ≥ 200 mm at the northern basins. Overflow of Manol and Muga rivers and damage to roads and margins | ACA (2011) Gázquez et al. (2004) |
| | 31-10-1993 | Precipitation of 270 mm in 4 d at the northern basins, leaving isolated villages, damaged roads, flooded housing and transport affections | ACA (2011) |
| SWP4 | 8-3-2010 | Heavy snowfall affecting 2/3 of the Catalan territory, with snow reaching the coastal municipalities | www.meteotecadecatalunya.cat (last accessed, April 2023) |
| | 26-12-2008 | Coastal storm with impacts mainly in the two northern basins with accompanying moderate rainfall with snow at low altitudes | Sanuy et al., (2021) Mendoza et al., (2011) Sánchez-Vidal et al., (2012) |
| | 9-4-2002 | Overflow of the Manol and Muga rivers (Girona North basin) | ACA (2011) |
| SWP5 | 3-11-2015 | Intense to severe rainfalls over the whole territory, which exceeded 200 mm in the internal basins causing 4 casualties due to flash flood events. On the coast, several flash floods were registered without casualties | www.meteotecadecatalunya.cat (last accessed, April 2023) |
| | 18-10-2003 | Some inundations and affections to roads at the upper sections of the Ter River. Moderate storms in coastal basins | ACA (2011) |
| | 26/27-9-1992 | Overflow of the Fluvià river and landslides in some interior basins | ACA (2011) |
| SWP2 | 29/30-11-2014 | P24 of 336 mm registered at the Ebro delta area and 200 mm at the northern basin. Intense rainstorms in all other areas, with accompanying moderate coastal storm | www.meteotecadecatalunya.cat (last accessed, April 2023) |
| | 15/16-3-2011 | Precipitations reaching more than 300 mm in 3 days at Viladrau, and exceeding 150 mm at multiple locations. Northern rivers such as Fluvià or Muga were close to overflowing (the emergency inundation plan was activated) | ACA (2011) |
| | 16-4-2004 | The heavy rainstorms that affected the north of Catalonia caused flooding in several rivers in the Girona northern and southern basins. | ACA (2011) |
| SWP6 | 21/22-1-2017 | Event featured by an intense wave storm affecting the north and central coasts, with rains almost reaching 200 mm at the northern basins | www.meteotecadecatalunya.cat (last accessed, April 2023) |
| SWP7 | 21-10-2000 | Overflow of two streams south of Tarragona, affecting multiple municipalities and | ACA (2011) |

| | | the cutting off local roads | |
|---|---|---|---|

The literature on compound coastal flooding has extensively explored the combined effects of surges with river discharges or surges with precipitation (Couasnon et al., 2018; Paprotny et al., 2018; Hendry et al., 2019; Camus et al., 2022; Whal et al., 2015; Wu et al., 2018; Bevacqua et al., 2019). While precipitation can serve as a proxy for hydrometeorological flooding

potential in small river catchments (Bevacqua et al., 2019), especially in the context of NW Mediterranean conditions, it's worth noting that surges play a relatively smaller role compared to wind waves in contributing to flooding in the area (Mendoza and Jiménez, 2009; Sanuy et al., 2020). Therefore, the analysis of significant Hs and wave power during coastal storms becomes essential to characterise the hazardous maritime component and "indicate" both coastal erosion and flooding potential.

Most of the research on compound coastal events has primarily focused on multivariate events at large scales due to the possibility of synergistic effects between hazards at sensitive locations (Whal et al., 2015; Couasnon et al., 2018; Paprotny et al., 2018; Wu et al., 2018; Bevacqua et al., 2019; Hendry et al., 2019; Camus et al., 2022). However, the recent impact of storm Gloria in January 2020 in the NW Mediterranean (e.g. Amores et al., 2020; Canals and Miranda, 2020) vividly demonstrated the accumulative impact of terrestrial and maritime hazards on various parts of the region. Under such conditions, civil

protection services had to prioritise interventions based on urgency because of the simultaneous occurrence of a large number of emergency calls (approximately 15,000) and required interventions (approximately 2,500) in all coastal sectors, as reported by the Ministry of Home Affairs of the Government of Catalonia. Understanding the characteristics of compound events, including their regional prevalence in terms of multivariate phenomena and the probabilities of synergistic versus cumulative potential impacts at local scale, is critical for enhancing event preparedness within the constraint of limited resources. This

consideration is particularly relevant for compound-type events where rainfall and coastal extremes occur in different basins. We successfully differentiated the pattern responsible for region-wide multivariate events (SWP1, Figure 7) from those that exhibit localized multivariate impacts, both with and without compounding waves or rain in other areas of the region. This underscores the effectiveness of the final classification in distinguishing impact distributions. Hence, the framework could be tested in the future across broader domains (e.g. the whole Spanish Mediterranean coast), provided that relevant data is

accessible.

The prediction skill of the BN played a pivotal role in selecting the optimal combination of atmospheric variables, domain size, and cluster numbers to effectively replicate rainfall and coastal storm intensity at the local scale. By incorporating seasonality and NAO to account for inter-annual and within-pattern variations, we achieved skill scores exceeding 0.4, signifying strong performance following the criteria of Sutherland et al (2004). The relative importance of season and NAO in

influencing wave heights and rainfall variations aligns with existing literature (Casanueva et al., 2014; Krichak et al., 2014; Criado-Aldanueva and Soto-Navarro, 2020; Morales-Márquez et al., 2020). Notably, superior performance was consistently observed in smaller domains for rain events, as described by Beck et al. (2016), while this relationship was less clear for wave events. Remarkably, the highest skill levels were consistently attained in the largest domain for compound events.

Our weighted skills, which consider how the BN captures uncertainty, reached excellent values exceeding 0.9. This further

validates the derived SWPs as a reliable reference baseline for classifying future dates and exploring shifts in the probabilities of compound severe conditions, along with their regional distribution, under climate change scenarios. Future enhancements may involve exploring additional atmospheric variables such as relative and specific humidity (see e.g. Teegavarapu et al., 2018), additional teleconnections (Casanueva et al., 2014; Taibi et al., 2019), or potentially influencing parameters such as NW Mediterranean Sea moisture and temperature before SWP occurrence (Rainaud et al., 2017). This represents a crucial step

towards characterizing the impact potential of the hazard drivers under study, making the methodology applicable as a rapid Early Warning System preceding the use of numerical models based on synoptic forecasting maps.

## 6. Conclusions

In the NW Mediterranean, where flash floods and coastal storms often coincide, we have successfully identified and characterized Synoptic Weather Patterns conducive to compound extreme events. To do this, we have developed a methodological framework that combines an objective synoptic classification method with a Bayesian Network. The BN played a pivotal role in selecting the optimal atmospheric variables, domain dimensions, and clusters, effectively capturing local-scale variations in key variables (daily rainfall and wave heights), thereby attaining robust skill scores. Furthermore, the analysis confirmed seasonality and the NAO as influential factors shaping wave heights and rainfall variations. It is noteworthy that, in the case of rainstorms, smaller domains consistently outperformed larger ones, while the largest domain consistently achieved the highest skill levels for compound events.

Identified SWPs conducive to compound events differ from those associated with extreme individual rain or wave events and typically involve upper-level lows and trough structures in combination with Mediterranean cyclones. These atmospheric configurations lead to severe to extreme coastal storms combined with convective systems.

The outstanding weighted skills obtained by our framework indicate its potential as a rapid early warning system for evaluating the likelihood of severe and extreme rain-wave conditions. This has the potential to enhance risk mitigation and emergency preparedness efforts. Moreover, the derived SWPs serve as validated baselines in terms of synoptic skill, enabling the classification of future events and the evaluation of compound events in the context of climate change. Future research directions may encompass the exploration of additional atmospheric variables and parameters, facilitating a deeper understanding of the role of potential drivers for these hazards.

## Author contributions

MS conceptualised the research, built the BN, performed the analysis, and co-wrote the manuscript. JCP conceptualised the research, guided the SWPs classification, and characterised the obtained types. SA developed the reconstruction of wave observations. JAJ conceptualised the research, co-wrote the manuscript, managed the project, and secured funding. All authors contributed to the interpretation and discussion of the obtained results and to the writing of the final manuscript.

## Competing interests

The contact author has declared that none of the authors has any competing interests.

## Acknowledgements

This study was conducted within the framework of the C3RiskMed (PID2020-113638RB-C21, AEI/10.13039/501100011033) research project, funded by the Spanish Ministry of Science and Innovation. The first author was funded by the Margarita Salas postdoc grant.

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
