# Peer review of "Synoptic weather patterns conducive to compound extreme rainfallwave events in the NW Mediterranean"

_Hydrology and Earth System Sciences, 2023_

## Author Comment (AC1)

**#1 - Reviewer**

The paper analyses weather events inflicting hazardous impacts over the Spanish northwest (NW) Mediterranean, such as floods and coastal storms characterized by high waves. The article analyses synoptic weather patterns (SWPs) conducive to compound events by combining an objective synoptic classification method based on principal component analysis and k-means clustering with Bayesian Networks (BNs). As the first method is a rather traditional method used in classifying synoptic patterns, the main innovation is adding BNs analysis. By adding BNs skills analysis to their classification method, the authors claim its advantage is characterizing the nonlinear relationship between SWPs and different variables for predicting compound extremes. The subdivision and research were done to contribute to understanding compound terrestrial-maritime phenomena in the study area and to assist in developing predictive and effective risk management strategies.

Dear Authors,

Your research is innovative by adding BNs to traditional methods. Your combined methodological framework shows promising results. However, my main issues are the work presentation, which is sometimes difficult to interpret, and your classification procedure. For example, the number of clusters you choose to describe the atmosphere seems too large. I.e., 18 weather types to describe 112 compound events? I have included my comments and suggestions below for you.

We appreciate the comments and time dedicated to the review and are very grateful for the accuracy of the observations. The revised version of the manuscript will address the two main issues pointed out by the reviewer, formal ones (work presentation) and the classification procedure. After considering the feedback from both reviewers, we are updating the manuscript. The revised version will now include an analysis of the model skill based on various factors: (i) different combinations of variables; (ii) domains; and (iii) number of clusters of the KMEANs approach "N". Using the BN model skill results (synoptic skill) as a criterion, we will select the appropriate combination of domain, variables and N. We have nearly completed these tests, and preliminary results indicate that the number of clusters can be reduced for all types of events without a significant loss of model skill.

We will enhance the manuscript by emphasizing the BN results, both in methodology and results section. As a result, the description of weather types will be condensed and simplified. Additionally, we will improve the quality of figures, add corresponding label, and enhance figure captions in both the main manuscript and supplementary material.

**Abstract**

1. Line 26 – What do you mean by 'reasonably'? Please give some quantitative estimates.
Quantitative estimates will be given in the revised version of the manuscript, for the final combination of domain, variables and N.

**Introduction**

2. Line 56 – A few recent review articles on extreme weather in the Mediterranean region are missing from your reference list.

Flaounas, E., Davolio, S., Raveh-Rubin, S., Pantillon, F., Miglietta, M. M., Gaertner, M. A., Hatzaki, M., Homar, V., Khodayar, S., Korres, G., Kotroni, V., Kushta, J., Reale, M., and Ricard, D.: Mediterranean cyclones: current knowledge and open questions on dynamics, prediction, climatology and impacts, Weather Clim. Dynam., 3, 173–208, https://doi.org/10.5194/wcd-3-173-2022 , 2022.

Hochman A, Marra F, Messori G, Pinto JG, Raveh-Rubin S, Yosef I, Zittis G,. 2022. Extreme weather and societal impacts in the Eastern Mediterranean. Earth System Dynamics 13(2): 749-777. https://doi.org/10.5194/esd-2021-55

Zittis, G., Almazroui, M., Alpert, P., Ciais, P., Cramer, W., Dahdal, Y., et al. (2022). Climate change and weather extremes in the Eastern Mediterranean and Middle East. Reviews of Geophysics, 60, e2021RG000762. https://doi.org/10.1029/2021RG000762

Thanks for the suggested references. They will be properly cited in the Introduction

3.    Line 61 – Do you mean 'objective' rather than 'subjective'?
Yes, indeed. This will be corrected.

4.    Lines 56 – 62 - A few articles on synoptic weather classification and their physical grounding are missing from your reference list. Please consider adding them.

For example:
The special issue entitled: Circulation-type classifications in Europe: results of the COST 733 Action .I would mention COST733 and describe its contributions in the introduction.

*Hochman A, Messori G, Quinting J, Pinto JG, Grams C. 2021. Do Atlantic-European weather regimes physically exist? Geophysical Research Letters 48: e2021GL095574. https://doi.org/10.1029/2021GL095574*
Thanks for the suggested references. They will be included in the Introduction

**Data**

5.    Lines 120 – 125 – Please add more information on how wave height reconstruction was done.
The following paragraph will be added to the section: " The reconstruction process involved a multilinear regression technique, using five oceanic variables (significant wave height, total wave mean period, mean wave period based on the first moment, mean zero-crossing wave period and total wave peak period) and three atmospheric variables (mean sea level pressure, wind speed and wind direction at 10m) as predictors for the targeted buoy variables (Hs or Tp). To consider the influence of wind and the morphology of the Catalan coast, the data was divided into four groups based on wind direction (0º to 90º, 90º to 180º, 180º to 270º, and 270º to 360º), resulting in distinct regression coefficients for each group."

6.    Line 130 – Please add latitude and longitude to Figure 1.
Figure will be corrected in the updated version.

7.    Figure 1 – Please increase the fonts of country labels. Please add the topography to the map.
This will be corrected in the updated version.

**Methods**

8. Line 146 – Please add detailed information in the caption of Figure 2 for the reader to be able to interpret your framework without looking it up in the main text. This comment can be applied to most of your figures.
Figure captions will undergo a thorough review following suggestions from both reviewers.

9. Line 161 – Typo remove 'as.'
This will be corrected in the updated version.

10. In Line 185 and throughout the text, I think you mean 'trough' rather than 'through.'
This will be corrected in the updated version.

11. Line 205 – Please add more information in the Figure 3 caption for the reader to be able to interpret without looking it up in the main text.
Figure captions will undergo a thorough review following suggestions from both reviewers.

12. Line 236 – Add information in the Figure 4 caption for the reader to be able to interpret without looking it up in the main text.
Figure captions will undergo a thorough review following suggestions from both reviewers.

**Results**

13. Line 246 – remove 'affected.'
This will be corrected in the updated version.

14. Line 264 – Are 18 weather types for 112 events too large? How many clusters do you have, and what is the explained variance? The issue of selecting a priori number of groups is essential, and you should discuss it.
In the updated version of the manuscript, the BN-model skill will be used to determine the optimal number of clusters, comparing various classification domains and variables.
For compound and wave-only events, N values of 6,10 and 14 are under exploration, while for rain-only events, N values of 10, 18 and 26 will be considered. These different N values correspond to the varying number of each event type.

*For example:*
*Falkena, S. K., de Wiljes, J., Weisheimer, A., Shepherd, T. G. (2020), Revisiting the identification of wintertime atmospheric circulation regimes in the Euro-Atlantic sector. Quarterly Journal of the Royal Meteorological Society, 146, 2801–2814. https://doi.org/10.1002/qj.3818*
Thanks for the suggested reference.

15. Section 4.2 – I suggest significantly reducing the text amount in this section. It isn't easy to read.
In response to the suggestions of both reviewers, this section will be restructured. The updated manuscript will place greater emphasis on the BN-skill analysis for the final classification and the posterior BN-characterization of weather types at the study site. Additionally, the description of weather types will be condensed in both the manuscript and the supplementary material.

16. Table 3 – Why is this table included in the text? Is it necessary, or can it be moved to the supplementary information?
This table will be moved to the  supplementary material.

17.    Figures 5, 6, and 7 – Please consider using anomalies in the figures rather than absolute values.
This point will be considered.

18.    In all figures, please use letters (a, b, c..) for each panel so the reader knows what panel you are referring to in the text. Also, you mention these letters in the captions of the figures, but they need to be shown on the figure.
Figure captions and quality will be improved (and corresponding labels/letters will be added to the panels) both in the manuscript and in the supplementary material.

**Discussion**

19.    The discussion section can be significantly shorter.
The updated discussion section will be reduced an adapted to the new version of the manuscript.

**Supporting information**

20.    There are too many panels in the supporting information figures, which could be clearer to interpret.
Supporting material will be adapted to the new version of the manuscript.

---

## Author Comment (AC2)

**#2 - Reviewer**

**General comments**

The paper by Sanuy et al. "Synoptic weather patterns conducive to compound extreme rainfall-wave events in the NW Mediterranean" presents (1) a synoptic-climatological assessment of compound extreme events over the NE part of Spain coupled with (2) the use of Bayesian networks that quantify the nonlinear links between the synoptic types and various variables describing the extreme events.

Overall, I find the study interesting and potentially worthy of publication, but only once the authors have successfully addressed the major issues. These issues comprise the methodology, the balance between the weather typing and its subsequent application, and the clarity of presentation (both text and figures).
We appreciate the comments and time dedicated to the review and are very grateful for the accuracy of the observations. The main issues pointed out by the reviewer will be addressed in the updated version of the manuscript.

After considering the feedback from both reviewers, we are updating the manuscript. The revised version will now include an analysis of the model skill based on various factors: (i) different combinations of variables; (ii) domains; and (iii) number of clusters of the KMEANs approach "N". Using the BN model skill results (synoptic skill) as a criterion, we will select the appropriate combination of domain, variables and N. We have nearly completed these tests, and preliminary results indicate that the number of clusters can be reduced for all types of events without a significant loss of model skill.

We will enhance the manuscript by emphasizing the BN results, both in methodology and results section. As a result, the description of weather types will be condensed and simplified. Additionally, we will improve the quality of figures, add corresponding label, and enhance figure captions in both the main manuscript and supplementary material.

**Specific comments**

In synoptic climatology, it is well known that links between synoptic-scale circulation and any conditioned surface variable are sensitive to how the circulation domain is defined, both in terms of its size and localization relative to one's area of interest. Based on the presented results, I do not understand why for studying extremes mainly dependent on close-by lows or troughs the authors decided to analyse atmospheric circulation over such a broad area. This is particularly striking for precipitation variables, which require smaller domains for good skill (Beck et al. 2016; IJC 36:7). This issue demonstrates itself when the authors train their networks, the skill of which is very low for precipitation. The authors suggest that including other variables including large-scale teleconnections may help, but I suggest that their primary focus be on smaller rather than larger scales. I strongly recommend that the authors experiment with the size and location (i.e. centre versus off-centre) of their circulation domain and assess the sensitivity of the networks' skill to these changes. If classifications are trained independently on each type of event, there is even no reason to use an identical region for each of them.

On a similar note, I am not convinced that including MSLP, Z500 and 10m wind components adds in this particular case any synoptic skill. The authors claim they were motivated by Miró et al. (2018), who however did not analyse events related to this analysis, as the authors claim in 344, but rather cold-air pools in which decoupling between local (site) and regional circulation systems was crucial. As part of discussion, the authors should include a sensitivity study showing how their BN outputs respond to inclusion of multiple mutually strongly correlated circulation variables.

The revised manuscript will feature an analysis of model skill based on different variable combinations, domains and number of clusters of the KMEANs approach "N".

The BN model skill, serving as a proxy for synoptic skill, will guide the selection of a suitable combination of domain, variables and N. Preliminary results indicate an improvement in model skill for some of the reduced domains, but not for all event types.

The analysis of atmospheric variables reveals that a chosen set (mslp, z500, u10, v10) demonstrates robust model skill across event types and interest variables at the local scale (both Hs, and P24h, for compound or uni-variate extremes) in some domains, although (mslp, z500, u) is also showing good performance. Depending on the results of the remaining tests, the chosen variable set will also be used in the updated version of the manuscript to classify the extreme events, or changed by the better alternative.

Based on the description, it is not clear whether the circulation variables were standardized prior to PCA decomposition – the authors only mention they used anomalies, which would not be enough to account for the differences in variables' variance. In such a case, only one variable would have a dominant impact on the classification output anyway.

Variables were standardized. This will be indicated in the updated manuscript.

This relates to my other comment. I do not understand why were the weather types defined independently for each of the extremes' types, why the authors decided to define that many types (I do not claim that it is wrong but – again – no evaluation of the effect on the networks' skill was carried out), especially considering the fact that each classification was subsequently (manually) clustered into three "supertypes" that have long been known to link to the studied extremes in the region.

In the study, classifications of all extreme days were initially tested before trying independent classifications for each type of the extreme.. The obtained skills were found to be higher when dividing the data per event type, resulting in a lower total number of weather types.

In the updated version of the manuscript, the BN-model skill will be used to determine the optimal number of clusters, comparing various classification domains and variables.

For compound and wave-only events, N values of 6,10 and 14 are under exploration, while for rain-only events, N values of 10, 18 and 26 will be considered. These different N values correspond to the varying number of each event type.

Compared with the synoptic types, the description of which seems unnecessarily too lengthy and overly detailed to me, the text describing the networks seems way too short. Note that the classification and its descriptive analysis represent an interesting exercise. However, the added value consists (or, should consist) in the subsequent analysis.

The BN skill will serve as a metric to guide the selection of variables, the number of clusters, and domains. Subsequently, it will be used to evaluate the predictive potential of different accompanying variables and describe the final classifications across the territory. In the updated manuscript, greater emphasis will be placed on the BN part of the method and results, while reducing the description of the synoptic types.

I strongly suggest decreasing the number of abbreviations used in the text. For instance, why using C/SR/SW instead of simply compound, rain and waves? In some parts, where these are combined with abbreviations of variables and types, the text is extremely hard to read. Last, please try to simplify your terminology, better explain it, and use it consistently.

This will be addressed in the new version of the manuscript.

The quality of the figures is not great. It is practically impossible to see the background maps on screen, let alone when printed.
Figure captions and quality will be improved (and corresponding labels will be added to the figure panels).

**Technical corrections/queries**

in what sense are PCA- or CA-based classifications "more subjective than those" mentioned above?
We actually meant to say "objective". This will be corrected in the updated manuscript.

One may argue that the study by Sanuy et al. (2021, HESS) already did this, albeit to a limited extent
The reference will be added to the introduction.

It is not clear what "different mechanisms" mean; different to what?
The authors concluded that the mechanisms in the NE Iberian Peninsula are different from those of the other sectors. This will be clearer in the updated manuscript.

74-76 Please reword this sentence, it is hard to understand
The sentence will be rephrased.

did you mean "forcing on"?
Yes, this will be corrected.

2.1 Consider moving 89-93 (local scale) after the description of the larger-scale factors of extreme events
This will be considered in the updated manuscript.

128-129 why was MSLP abbreviation defined in 82 but z500 was not?
Abbreviations will be consistently defined at first appearance.

Are waves also a meteorological driver or is it a typo?
Yes, we refer to meteo-ocean drivers. .

I do not understand what "weather classifications associated SWP" means. Aren't SWP a synonym for weather (circulation) classification/types?
We will rephrase the sentence to: "Weather classification: obtaining SWP for each event type"

Isn't the classification method used to identify dominant SWPs? Also, what is "dominant", "critical", "target variables", and "SWP system"? I like the inclusion of the general framework and Fig. 2, but at this point they are not very clear, i.a. due to inclusion of abbreviations that have not been defined
Both the figure caption and the manuscript text referring to the General Framework figure will be modified based on reviewers' comments.

Was P24h explained in Sect. 2?
The abbreviation will be introduced in section 2.2.1.

P24 and P24h: what is the difference?
It is the same, P24. We will check for consistent usage throughout the manuscript.

"as as"
The typo will be corrected.

Does "separately" mean that each variable's anomaly fields were decomposed separately?
Yes. This will be clearer in the updated version.

What do you mean by "fundamental" modes and how does they relate to the anomalies?
It was a way of describing what the PCA does with a given dataset. The phrase will be removed.

So how many PCs/clusters were finally selected?
In the updated manuscript, this section will be modified, and the Knee-test will no longer be used to determine the number of clusters. Instead, an analysis of the BN-skill will be utilized. The final classification will specify the number of principal components (PC) and clusters.

How were they grouped? You need to show the patterns and refer to them from here. Also this text suggests that you join the types in three final types for which you present results, which is not the case. Please reword. Furthermore, I recommend using a different term for your overarching three patterns, such as "supertypes", to clearly distinguish them from SWPs.
The "classification" in "supertypes" will undergo a deep review. Instead, the structures observed in the most severe weather types will be described (i.e. presence of Mediterranean Cyclones, presence of Cut-Offs, etc.).

all?
It is a typo. The sentence will be rephrased to "linear relationships between all variables"

what is parent?
In this context, synonym of predictor, and opposite to target. This will be clearer in the updated version.

is "SWP system's proficiency" the same as "classification's ability"? If the term system is important, please explain it sooner.
The term system is not important. We will use classification's ability.

correct "(SWPs)"
This will be corrected.

Why are not all of these parameters/variables mentioned in Sect.2?
Hs, P24, Tp and wave direction are presented in Sect.2 (they are raw data). Wave power content is calculated from Hs and Tp as explained in Section 3 (is one of the first methodological steps in event identification). The season of the events is also retained (this will be included in section 3, as it is not mentioned).

Standard deviation of the skill?
Standard deviation of the predictions.

Why do you repeatedly define abbreviations that you then don't even consistently use?
Abbreviations related to the event types will be dropped.

Please add panel captions into the figures, and refer directly to individual panels from the text as much as possible.
This will be done in the updated version of the manuscript.

How is there a decrease from North to South is the values are identical for the central and southern regions?
The sentence will be rephrased.

What does "their" refer to?
The sentence will be rephrased to "The weather types were linked to the event season"

What was verified?
The sentence will be rephrased.

The sentence "highlighting...severity" makes no sense
The sentence will be rephrased.

Are local multivariate events the same as compound events? I am becoming lost in the terminology.
A compound event may affect a basin as multivariate (both components exceed the extreme threshold in the same basin), while affecting other basins with only one of the components. Likewise, a compound event may not be multivariate at any basin, but compound on a regional scale. We will simplify the description of the weather types, focusing on the intensity and probability of local spatial co-occurrence. In any case, we will try to be clearer when talking about the different types of compound events to avoid confusion and "noise".

"The dominant..." > "The two dominant..." ... "all of them" means those in Figs 5+6 or something else? Please explain/reword "minimum relative cumulative spatial temporal all basins probability"
This will be rephrased in the updated manuscript.

What do you mean by "unstable conditions"?
This will be rephrased in the updated manuscript.

In all seven? I can hardly check because of the quality of the map, but it seems unlikely that all seven types (than even do not belong to the same "supertype") have the same feature.
All seven types present wind fields over the Mediterranean characterized by long trajectories over the water from E to W.

Please explain the abbreviations of types in the text, or use generic names (e.g., type1, etc.). It is impossible to remember which type is which.
Abbreviations will be dropped, as the description of weather types will be simplified.

There are more objective alternatives to visual checking for similarity. One can calculate a pattern-correlation matrix, or a distance matrix, project all SWPs to the first principal component plane, or alternatively one can use e.g. Sammon mapping to test whether SPWs that you identify with different supertypes truly occupy distinct parts of the data space without the linear/orthogonal PCA constraints.
The "classification" in "supertypes" will undergo a deep review. Instead, the structures observed in the most severe weather types will be described (i.e. presence of Mediterranean Cyclones, presence of Cut-Offs, etc.).

is it the same skill as in 221? I suggest adding a reference to the section where it is explained
Yes. This will be done in the new version remove comma before "leading". Better explain "dominant" and "general"
This will be addressed in the updated version.

424, 737 etc. What is "weather configuration"? Please select a clear terminology and use it consistently.
This will be addressed in the updated version.

This describes the results of the reference, or yours?
Ours. The discussion section will undergo a deep review, adapted to the new version of the manuscript, and all minor comments will be addressed.

consider changing "this work" to "their work" or similar
The discussion section will undergo a deep review, adapted to the new version of the manuscript, and all minor comments will be addressed.

---

## Author Response (AR1)

**Review of the manuscript: Synoptic weather patterns conducive to compound extreme rainfall-wave events in the NW Mediterranean by Marc Sanuy, Juan C. Peña, Sotiris Assimenidis, and José A. Jiménez**

**Introductory letter to reviewers**

Dear reviewers,

We appreciate the comments and time dedicated to the review and are very grateful for the accuracy of the observations of both reviewers. We truly believe that they helped improving significantly our work and the way it is presented. The main issues raised by both reviewers can be summarized as follows:

1. Methodological concerns. Both reviewers highlighted concerns about the number of clusters used to classify compound events. The second reviewer also pointed out issues related to the spatial dimensions of the classification domain and the choice of atmospheric variables.

2. Results section structure. The reviewers' feedback pointed out an imbalance in the weight of the results' subsection. Specifically, the description of the synoptic types was deemed lengthy and complex due to additional manual classification in three main configurations, and an overuse of abbreviations. In contrast, the innovative part of our methodological framework, the use of the Bayesian Network (BN) coupled with an objective classification, received less attention. Some BN results were also mixed with the description of the Synoptic Weather Patterns (SWPs).

3. Presentation quality. Concerns were raised regarding the quality of figures, figure captions and the aforementioned excessive use of abbreviations throughout the manuscript.

As a result of this feedback, both the study and the manuscript have undergone substantial revisions. These changes involved modifying the methodological framework and reorganizing the results section. In the updated study and manuscript, we integrated the BN into the methodological process to facilitate the selection of an optimal combination of domain, classification variables and number of clusters based on synoptic skill. Therefore, entire subsections were completely rewritten (highlighted in color in the annotated version of the manuscript. To address the three main points raised above, the following updates were made:

**Methodological enhancements**

1. We expanded the analysis by testing different combinations of atmospheric variables, including mslp, z500, u, v and the new addition of t850. These variables were tested across 7 different domains, varying in size and location relative to the study area (centered and off-center). The division between individual rain/wave and compound events was maintained, while also exploring the impact of varying the number of clusters (with values of 6, 10, 14, 18 and 26).

2. The BN is now used to evaluate the tested classifications in terms of BN-skill, a measure of synoptic skill, in predicting the target variables Hs and P24. Additionally, we introduced the monthly mean NAO as an extra variable and assessed its impact on predicting skill when used alongside SWPs.

**Results section structure**

3. In the updated results section 4.2, the BN analysis precedes the description of SWPs (section 4.3). The SWP description was completely revised, with the removal of the additional classification into 3 main configurations and the historical event's text, which is now presented in a table in the discussion section. The description of SWPs is now based solely on their probabilities of occurrence conditioned to pre-defined levels of intensity at regional scale and their regional distributions of impacting drivers. Both probability distributions were obtained using the BN, and their interpretation is now more comprehensively explained in the corresponding methodological section (section 3.4).

**Improved presentation**

4. We have enhanced figure quality and added panels where suggested. Figure captions have been revised to provide more comprehensive details, making figure interpretation independent of the main text. All abbreviations related to event types and specific weather patterns have been removed, resulting in improved overall readability. These improvements also extended to supplementary figures, which have been reduced in number.

In what follows, we provide detailed responses to each of the comments and questions raised by the reviewers.

Please be aware that some of the reviewers' style/grammar recommendations have become obsolete as parts of the text have been fully rewritten. Where they are still applicable (the paragraph has not been rewritten, they have been implemented).

**#1 - Reviewer**

The paper analyses weather events inflicting hazardous impacts over the Spanish northwest (NW) Mediterranean, such as floods and coastal storms characterized by high waves. The article analyses synoptic weather patterns (SWPs) conducive to compound events by combining an objective synoptic classification method based on principal component analysis and k-means clustering with Bayesian Networks (BNs). As the first method is a rather traditional method used in classifying synoptic patterns, the main innovation is adding BNs analysis. By adding BNs skills analysis to their classification method, the authors claim its advantage is characterizing the nonlinear relationship between SWPs and different variables for predicting compound extremes. The subdivision and research were done to contribute to understanding compound terrestrial-maritime phenomena in the study area and to assist in developing predictive and effective risk management strategies.

Dear Authors,
Your research is innovative by adding BNs to traditional methods. Your combined methodological framework shows promising results. However, my main issues are the work presentation, which is sometimes difficult to interpret, and your classification procedure. For example, the number of clusters you choose to describe the atmosphere seems too large. I.e., 18 weather types to describe 112 compound events? I have included my comments and suggestions below for you.
See Introductory letter to reviewers

**Abstract**
1. Line 26 – What do you mean by 'reasonably'? Please give some quantitative estimates.
In the new version, the quantitative estimate of the obtained skill is given in the Abstract.

**Introduction**

2. Line 56 – A few recent review articles on extreme weather in the Mediterranean region are missing from your reference list.

Flaounas, E., Davolio, S., Raveh-Rubin, S., Pantillon, F., Miglietta, M. M., Gaertner, M. A., Hatzaki, M., Homar, V., Khodayar, S., Korres, G., Kotroni, V., Kushta, J., Reale, M., and Ricard, D.: Mediterranean cyclones: current knowledge and open questions on dynamics, prediction, climatology and impacts, Weather Clim. Dynam., 3, 173–208, https://doi.org/10.5194/wcd-3-173-2022 , 2022.

Hochman A, Marra F, Messori G, Pinto JG, Raveh-Rubin S, Yosef I, Zittis G,. 2022. Extreme weather and societal impacts in the Eastern Mediterranean. Earth System Dynamics 13(2): 749-777. https://doi.org/10.5194/esd-2021-55

Zittis, G., Almazroui, M., Alpert, P., Ciais, P., Cramer, W., Dahdal, Y., et al. (2022). Climate change and weather extremes in the Eastern Mediterranean and Middle East. Reviews of Geophysics, 60, e2021RG000762. https://doi.org/10.1029/2021RG000762

Among these suggested references, the one referred to Mediterranean cyclones was included in the Introduction section.

3. Line 61 – Do you mean 'objective' rather than 'subjective'?
Yes, indeed. This was corrected.

4. Lines 56 – 62 - A few articles on synoptic weather classification and their physical grounding are missing from your reference list. Please consider adding them.
For example:
The special issue entitled: Circulation-type classifications in Europe: results of the COST 733 Action. I would mention COST733 and describe its contributions in the introduction.
*Hochman A, Messori G, Quinting J, Pinto JG, Grams C. 2021. Do Atlantic-European weather regimes physically exist? Geophysical Research Letters 48: e2021GL095574. https://doi.org/10.1029/2021GL095574*

Thank you. A reference to the COST733 project and its main contributions have been included in the introduction. The new paragraph reads as follows (line 62):

*"The EU COST 733 project (Huth et al., 2010) has significantly contributed to advancing scalable classification techniques applicable to various European regions. Several classification methodologies were proposed and rigorously compared within this project, highlighting that different classification approaches demonstrated comparable effectiveness (e.g., Philip et al., 2010, Beck and Philipp, 2010). Notably, the synoptic skill of weather classifications, i.e. their capacity to accurately replicate the magnitudes of key target variables at the local scale, was identified as particularly sensitive to various methodological aspects inherent to objective approaches. These encompassed factors such as the predefined number of classification groups, the selection of atmospheric variables and their number, and the spatial dimensions of the classification domain (see e.g. Philip et al., 2016; Beck et al., 2016; Teegavarapu et al., 2018; Falkena et al., 2020)"*

**Data**

5. Lines 120 – 125 – Please add more information on how wave height reconstruction was done.
The following paragraph has been added to the section (line 134):
*"The reconstruction process was based on ERA-5 data (SMC, 2021) and utilized a multilinear regression technique, employing five oceanic variables (significant wave height, total wave mean period, mean wave period based on the first moment, mean zero-crossing wave period and total wave peak period) and three atmospheric variables (mean sea level pressure, wind speed and wind direction at 10m) as predictors for the targeted buoy variables (Hs or Tp). To account for the influence of wind and the morphology of the Catalan coast, the data was categorized into four groups based on wind direction (0º to 90º, 90º to 180º, 180º to 270º, and 270º to 360º), resulting in distinct regression coefficients for each group"*

6. Line 130 – Please add latitude and longitude to Figure 1.
The updated Figure 1 includes the topography map, the main rivers, the latitude and longitude and increased font size of country labels.

7. Figure 1 – Please increase the fonts of country labels. Please add the topography to the map.
See previous response.

**Methods**
8. Line 146 – Please add detailed information in the caption of Figure 2 for the reader to be able to interpret your framework without looking it up in the main text. This comment can be applied to most of your figures.
All figure captions underwent a thorough review following suggestions from both reviewers.

9. Line 161 – Typo remove 'as.'
This part was rewritten to clarify the types within compound events and thus the typo does no longer exist.

10. In Line 185 and throughout the text, I think you mean 'trough' rather than 'through.'
This was corrected in the updated version of the manuscript.

11. Line 205 – Please add more information in the Figure 3 caption for the reader to be able to interpret without looking it up in the main text.
All figure captions underwent a thorough review following suggestions from both reviewers.

12. Line 236 – Add information in the Figure 4 caption for the reader to be able to interpret without looking it up in the main text.
All figure captions underwent a thorough review following suggestions from both reviewers.

**Results**
13. Line 246 – remove 'affected.'
This was corrected in the updated version.

14. Line 264 – Are 18 weather types for 112 events too large? How many clusters do you have, and what is the explained variance? The issue of selecting a priori number of groups is essential, and you should discuss it.
In the new version of the manuscript, the BN-model skill was used to determine the optimal number of clusters, comparing various classification domain sizes and variables.

For compound and wave-only events, N values of 6, 10 and 14 were explored, while for rain-only events, N values of 10, 18 and 26 were considered. In addition, classifications yielding groups with less than 5 dates were filtered out, considering them not robust.

With this criterion, compound events (112 dates) are now classified in 10 groups, wave events (74 dates) are also classified in 10 groups whereas rain events (376 dates) are classified in 26 groups.

The justification of such selection is presented in new section 4.2 and related supplementary material.

*For example:*

*Falkena, S. K., de Wiljes, J., Weisheimer, A., Shepherd, T. G. (2020), Revisiting the identification of wintertime atmospheric circulation regimes in the Euro-Atlantic sector. Quarterly Journal of the Royal Meteorological Society, 146, 2801–2814. https://doi.org/10.1002/qj.3818*

This reference was included in the Introduction section (see response to comment 4).

15.    Section 4.2 – I suggest significantly reducing the text amount in this section. It isn't easy to read.

In response to the suggestions by both reviewers, the results section was restructured. Now, section 4.2 presents the results of the BN-skill analysis, while section 4.3 presents the compound SWPs (old section 4.2). In the updated version, we now exclusively describe SWPs related to compound events, eliminating the use of the previous 3 main configurations (Cut-Off, Atlantic Low and Trough). This simplification extends to a reduction of the number of abbreviations. To maintain conciseness and focus on the core content, all text pertaining to historical events has been removed from Section 4.3. Instead, we have relocated the relevant information to a dedicated table in the Discussion section. Panels d) of the figures accompanying the SWP description have been simplified, which allows for more concise and efficient description.

16.    Table 3 – Why is this table included in the text? Is it necessary, or can it be moved to the supplementary information?

The table has been moved to the discussion section. We preferred to maintain it as it serves to illustrates the relevance of some recorded events and their associate SWP.

17.    Figures 5, 6, and 7 – Please consider using anomalies in the figures rather than absolute values.

The modified equivalent figures (section 4.3) now show the anomalies as suggested by the reviewer.

18.    In all figures, please use letters (a, b, c..) for each panel so the reader knows what panel you are referring to in the text. Also, you mention these letters in the captions of the figures, but they need to be shown on the figure.

Panel letters were included in all figures that required them. Specifically, for figures accompanying the SWPs' description panels refer to the column elements, as the SWP number serves to distinguish the rows. Also, accompanying figure captions are now more detailed, helping overall interpretation.

**Discussion**

19.    The discussion section can be significantly shorter.

The updated discussion section was adapted to the new version of the manuscript.

**Supporting information**

20.    There are too many panels in the supporting information figures, which could be clearer to interpret.

The number of supporting information figures was reduced, and those presenting SWPs that were not shown in the main manuscript have been simplified to only the severe and extreme cases. Figure panel letters were also included here, and corresponding figure captions improved.

**#2 - Reviewer**

**General comments**

The paper by Sanuy et al. "Synoptic weather patterns conducive to compound extreme rainfall-wave events in the NW Mediterranean" presents (1) a synoptic-climatological assessment of compound extreme events over the NE part of Spain coupled with (2) the use of Bayesian networks that quantify the nonlinear links between the synoptic types and various variables describing the extreme events.

Overall, I find the study interesting and potentially worthy of publication, but only once the authors have successfully addressed the major issues. These issues comprise the methodology, the balance between the weather typing and its subsequent application, and the clarity of presentation (both text and figures).
See Introductory letter to reviewers

**Specific comments**

In synoptic climatology, it is well known that links between synoptic-scale circulation and any conditioned surface variable are sensitive to how the circulation domain is defined, both in terms of its size and localization relative to one's area of interest. Based on the presented results, I do not understand why for studying extremes mainly dependent on close-by lows or troughs the authors decided to analyse atmospheric circulation over such a broad area. This is particularly striking for precipitation variables, which require smaller domains for good skill (Beck et al. 2016; IJC 36:7). This issue demonstrates itself when the authors train their networks, the skill of which is very low for precipitation. The authors suggest that including other variables including large-scale teleconnections may help, but I suggest that their primary focus be on smaller rather than larger scales. I strongly recommend that the authors experiment with the size and location (i.e. centre versus off-centre) of their circulation domain and assess the sensitivity of the networks' skill to these changes. If classifications are trained independently on each type of event, there is even no reason to use an identical region for each of them.

On a similar note, I am not convinced that including MSLP, Z500 and 10m wind components adds in this particular case any synoptic skill. The authors claim they were motivated by Miró et al. (2018), who however did not analyse events related to this analysis, as the authors claim in 344, but rather cold-air pools in which decoupling between local (site) and regional circulation systems was crucial. As part of discussion, the authors should include a sensitivity study showing how their BN outputs respond to inclusion of multiple mutually strongly correlated circulation variables.
The revised manuscript includes the analysis of model skill based on different variable combinations, domains and number of clusters of the k-means approach (N).

The BN model skill, serving as a proxy for synoptic skill, guided the selection of a suitable combination of domain, variables and N.

The main results after experimenting with domain size and location showed that the smallest domain exhibited superior performance in representing P24 for individual rain events. In contrast, larger domains proved more effective in capturing the relevant variables of interest for compound and wave events.

Also, after experimenting with different variable combinations, results showed that while previous combination (mslp, z500, u and v) was a reasonable choice for compound events, it was surpassed by the use of z500 and u in isolation. For rain and wave events, alternative variable combinations demonstrated better performance compared to the previous selection. As suggested by the reviewer, each event type was classified using an independent combination of domain, variables and N.

In the updated version, we introduced t850 as an atmospheric variable in the tested combinations. However, our findings indicated that the inclusion did not lead to improved results. Additionally, we incorporate the NAO to evaluate its influence on the synoptic skill of the classifications, together with the seasonality.

Based on the description, it is not clear whether the circulation variables were standardized prior to PCA decomposition – the authors only mention they used anomalies, which would not be enough to account for the differences in variables' variance. In such a case, only one variable would have a dominant impact on the classification output anyway.
Variables were standardized. This is now explicit in the updated manuscript.

This relates to my other comment. I do not understand why were the weather types defined independently for each of the extremes' types, why the authors decided to define that many types (I do not claim that it is wrong but – again – no evaluation of the effect on the networks' skill was carried out), especially considering the fact that each classification was subsequently (manually) clustered into three "supertypes" that have long been known to link to the studied extremes in the region.
Initially, classifications for all extreme days were performed collectively before exploring independent classifications for each type of the extreme event. When focusing on compound events, we found that dividing the data per event type led to higher skills and required a lower number of clusters. This tailored approach significantly enhanced the accuracy of the representations for compound events. In section 4.1, a preliminary analysis was conducted to compare compound with individual events. This analysis highlighted substantial differences in magnitude between these various event types, emphasizing the need for distinct classifications.

In the updated version, "supertypes" have been eliminated from the classification process. Instead, the BN-skill analysis lead to an objective definition of the number of SWPs.

Compared with the synoptic types, the description of which seems unnecessarily too lengthy and overly detailed to me, the text describing the networks seems way too short. Note that the classification and its descriptive analysis represent an interesting exercise. However, the added value consists (or, should consist) in the subsequent analysis.
The description of SWPs has undergone a substantial reduction in length, while still providing valuable information for interested readers. Reduction was mainly focused on the removal of text related to historical events, the elimination of "supertypes", and simplification in the use of abbreviations. Also, only SWPs of compound events are now presented (while rain and wave SWPs are only mentioned in the discussion and extreme cases presented in supplementary figures).
The text describing the networks, their training, and their interpretation has been expanded in the Methods section, while its added value is now more stressed in the Results section by its use to compare classifications, while at the same time providing SWPs characteristics and distribution of impacts across the territory

I strongly suggest decreasing the number of abbreviations used in the text. For instance, why using C/SR/SW instead of simply compound, rain and waves? In some parts, where these are combined with abbreviations of variables and types, the text is extremely hard to read. Last, please try to simplify your terminology, better explain it, and use it consistently.
Following reviewer's suggestion, *compound*, *rain* and *wave* events are now used, dropping the previous definition and abbreviations.

The quality of the figures is not great. It is practically impossible to see the background maps on screen, let alone when printed.
Figure captions and quality were improved overall.

**Technical corrections/queries**

61 in what sense are PCA- or CA-based classifications "more subjective than those" mentioned above?

We actually meant to say "objective". This was corrected in the updated manuscript.

71 One may argue that the study by Sanuy et al. (2021, HESS) already did this, albeit to a limited extent

We have rephrased the sentence to *"...synoptic weather patterns (SWPs) conducive to compound events involving both hazards in the region have not been thoroughly studied" (line 78)*

73 It is not clear what "different mechanisms" mean; different to what?

The authors concluded that the mechanisms in the NE Iberian Peninsula are different from those of the other sectors. The sentence now reads: *"... the results suggested that the climatology of extreme weather events in the NE sectors (NW Mediterranean coast) is driven by different mechanisms, relative to other sectors". (line 81)*

74-76 Please reword this sentence, it is hard to understand

The sentence was rephrased to *"The complex orography of the region influences precipitation and wind patterns, and the coastal storm climate is characterised by the waves playing a more significant role to erosion and flooding than surges (Mendoza and Jiménez, 2009; Sanuy et al., 2020)". (line 86)*

90 did you mean "forcing on"?

Yes, this was corrected.

2.1 Consider moving 89-93 (local scale) after the description of the larger-scale factors of extreme events

This part of the text was left without changes in structure.

128-129 why was MSLP abbreviation defined in 82 but z500 was not?

Abbreviations are now consistently defined at first appearance.

136 Are waves also a meteorological driver or is it a typo?

Yes, we refer to meteo-ocean drivers. The sentence now reads: *"(A) Event identification: This step involves identifying and characterizing rain and wave storm events as individual or compound events, based on the presence of either or both meteo-marine drivers." (line 152)*

137 I do not understand what "weather classifications associated SWP" means. Aren't SWP a synonym for weather (circulation) classification/types?

The sentence now changed to *"(B) Weather classifications: Here, synoptic weather patterns were identified by using a PCA and k-means approach, utilizing multiple atmospheric variables, domains, and varying cluster sizes".(line 149)*

138 Isn't the classification method used to identify dominant SWPs? Also, what is "dominant", "critical", "target variables", and "SWP system"? I like the inclusion of the general framework and Fig. 2, but at this point they are not very clear, i.a. due to inclusion of abbreviations that have not been defined

We dropped the use of "dominant" to refer to the most intense SWPs. Now they are referred to as "critical" in the Figure, and the different intensity levels to which the BN analysis is performed are defined in the BN section.

144 Was P24h explained in Sect. 2?

The abbreviation is now introduced at first appearance.

158 P24 and P24h: what is the difference?

It is the same, P24. We checked for consistent usage throughout the manuscript.

161 "as as"

The typo was corrected.

174 Does "separately" mean that each variable's anomaly fields were decomposed separately?

Yes. The sentence was rephrased to *"we applied PCA to the standardized anomalies of the maps, separately for each variable…"*. (line 208)

175 What do you mean by "fundamental" modes and how does they relate to the anomalies?

It was a way of describing what the PCA does with a given dataset. The phrase was removed.

179 So how many PCs/clusters were finally selected?

In the updated manuscript, this section was modified, and the Knee-test was no longer be used to determine the number of clusters.

Instead, an analysis of the BN-skill was utilized across different combinations of atmospheric variables, domains and number of clusters. These combinations, and the number of PCs used are now depicted in Table 2. The number of PCs corresponds to a 99% of explained variance in the case of variables with low variability (mslp, z500 and t850), and 90% of explained variance in the case of wind components (u, v) that feature a relatively larger variability.

181 How were they grouped? You need to show the patterns and refer to them from here. Also, this text suggests that you join the types in three final types for which you present results, which is not the case. Please reword. Furthermore, I recommend using a different term for your overarching three patterns, such as "supertypes", to clearly distinguish them from SWPs.

The "classification" in "supertypes" was eliminated in the updated version of the manuscript. Instead, SWPs are presented grouped by their probability of occurrence conditioned on the intensity levels defined in section (3.4). The structures such as presence of Mediterranean cyclones/lows, or presence of cut-offs / troughs are only mentioned in the pattern descriptions.

197 all?

The sentence has been rephrased to *"Descriptive BNs were used to see through the nonlinear relationships between variables in order to explore specific relationships between predictors and target variables"*. (line 220)

200 what is parent?

In this context, synonym of predictor, and opposite to target. This part of the manuscript was re-written and this word does not appear anymore.

201 is "SWP system's proficiency" the same as "classification's ability"? If the term system is important, please explain it sooner.

The term system is not important. Across the manuscript, we now use the term synoptic skill, or classification's ability to reproduce the variability of target variables.

204 correct "(SWPs)"

This part was canceled as we don't mention spare days in the new version of our work

209 Why are not all of these parameters/variables mentioned in Sect.2?

Hs, P24, Tp and wave direction are presented in Sect.2 (they are raw data). Wave power content is calculated from Hs and Tp as explained in Section 3 (is one of the first methodological steps in event

identification). The season and monthly average NAO corresponding to each event is also retained (this was now included in section 3.1, as it is was not mentioned).

223 Standard deviation of the skill?
Standard deviation of the predictions.

230 Why do you repeatedly define abbreviations that you then don't even consistently use?
Abbreviations related to the event types were dropped.

235 Please add panel captions into the figures, and refer directly to individual panels from the text as much as possible.
All figure captions underwent a thorough review following suggestions from both reviewers.

252 How is there a decrease from North to South is the values are identical for the central and southern regions?
The sentence was rephrased to: "*Notably, the annual frequency of compound events was highest at the north, averaging 2.6 events per year, compared to 1.2 events per year in the central and southern coastal sectors.*" (line 287)

263 What does "their" refer to?
The section (now section 4.3) was completely rewritten

266 What was verified?
The section (now section 4.3) was completely rewritten

266 The sentence "highlighting...severity" makes no sense
The section (now section 4.3) was completely rewritten.

273 Are local multivariate events the same as compound events? I am becoming lost in the terminology.
A compound event has the potential to impact a basin in various ways. It may exhibit a multivariate character, when both components exceed the extreme threshold within the same basin, while affecting other basins with only one of the components exceeding the threshold. Alternatively, a compound event may not be multivariate at any basin but can still have a compound effect on a regional scale. In the updated section 4.3 we have refined the description of the weather types, with a specific focus on the intensities and the probability of local spatial co-occurrence, which is indicative of multivariate characteristics. Additionally, we have rephrased the relevant part of the methodology section, specifically in Section 3.2 (line 189) were the different types of compound events are defined: "*Within the compound events, further classification at the basin scale was done, distinguishing between multivariate events (both rain and waves in the same basin), compounding wave events (basin affected only by waves), or compounding rain events (basin affected only by rain). Sanuy et al. (2021) emphasized the importance of this classification proposed by Zscheischler et al (2020) for risk management during compound events in NW Mediterranean conditions.*"

282 "The dominant..." > "The two dominant..." ... "all of them" means those in Figs 5+6 or something else? Please explain/reword "minimum relative cumulative spatial temporal all basins probability"
This text does not appear in the updated version of the manuscript.

285 What do you mean by "unstable conditions"?
The section (now section 4.3) was completely rewritten.

287 In all seven? I can hardly check because of the quality of the map, but it seems unlikely that all seven types (than even do not belong to the same "supertype") have the same feature.
All seven types presented wind fields over the Mediterranean characterized by long trajectories over the water from E to W. However, this text does not appear anymore as the section (now section 4.3) was completely rewritten.

297 Please explain the abbreviations of types in the text, or use generic names (e.g., type1, etc.). It is impossible to remember which type is which.
Abbreviations of SWP have been eliminated, as the description of weather types was simplified.

346 There are more objective alternatives to visual checking for similarity. One can calculate a pattern-correlation matrix, or a distance matrix, project all SWPs to the first principal component plane, or alternatively one can use e.g. Sammon mapping to test whether SWPs that you identify with different supertypes truly occupy distinct parts of the data space without the linear/orthogonal PCA constraints.
The "classification" in "supertypes" has been eliminated. Instead, the structures observed in the severe and extreme weather types will be described (i.e. presence of Mediterranean cyclones/lows, presence of cut-offs/troughs, etc.).

370 is it the same skill as in 221? I suggest adding a reference to the section where it is explained
Yes. This is now in section 4.2 and reference to the methodological section 3.4 is introduced.

393 remove comma before "leading". Better explain "dominant" and "general"
This part of the discussion was totally rewritten.

424, 737 etc. What is "weather configuration"? Please select a clear terminology and use it consistently.
It was referring to the "supertypes", which are now dropped.

429 This describes the results of the reference, or yours?
This paragraph was canceled from the discussion, as the selection of classification variable is now part of the methodological framework.

448 consider changing "this work" to "their work" or similar
This part was now changed to "*They found that the most extreme storms (class V events) occurred in the presence of a Mediterranean cyclone. Configurations without Mediterranean lows, referred to as Southern and Eastern Advections, were also associated with less severe coastal storms ranging from class II to IV.*" (line 468)